# Thermoreversible Gelation with Two-Component Mixed Cross-Link Junctions of Variable Multiplicity in Ternary Polymer Solutions

**DOI:** 10.3390/gels7030089

**Published:** 2021-07-11

**Authors:** Fumihiko Tanaka

**Affiliations:** Department of Polymer Chemistry, Graduate School of Engineering, Kyoto University, Katsura, Kyoto 615-8510, Japan; ftanaka@kmj.biglobe.ne.jp

**Keywords:** thermoreversible gelation, binary network, mixed cross-links, cross-link multiplicity, optimal gel point, reentrant sol-gel-sol transition, phase separation, spinodal line, hydrogen bonding, hydrophobic association, interpenetrating network

## Abstract

Theoretical scheme is developed to study thermoreversible gelation interfering with liquid–liquid phase separation in mixtures of reactive *f*-functional molecules R{A${}_{f}$} and *g*-functional ones R{B${}_{g}$} dissolved in a common solvent. Formed polymer networks are assumed to include multiple cross-link junctions containing arbitrary numbers $k_{1}$ and $k_{2}$ of functional groups A and B of each species. Sol-gel transition lines and spinodal lines are drawn on the ternary phase plane for some important models of multiple cross-link junctions with specified microscopic structure. It is shown that, if the cross-link structure satisfies a certain simple condition, there appears a special molar ratio of the two functional groups at which gelation takes place with a lowest concentration of the solute molecules, as has been often observed in the experiments. This optimal gelation concentration depends on *f* and *g* (functionality) of the solute molecules and the numbers $k_{1}$ and $k_{2}$ (multiplicity) of the functional groups in a cross-link junction. For cross-links which allow variable multiplicity, special attention is paid on the perfectly immiscible cross-links leading to interpenetrating polymer networks, and also on perfectly miscible cross-links leading to reentrant sol-gel-sol transition. Results are compared with recent observations on ion-binding polymer solutions, polymer solutions forming recognizable biomolecular complexes, polymer/surfactant mixtures, hydrogen-bonding polymers, and hydrophobically-modified amphiphilic water-soluble polymers.

## 1. Introduction: Binary Gels with Mixed Cross-Links

A majority of physical gels made up of natural polymers contain more than two component functional molecules and/or groups in their cross-link junctions [1,2,3,4,5,6]. Synthetic gels are also often made by mixing chemical cross-linkers, ions, other reactive molecules into polymer solutions [2,4,6]. More recently, dual polymer networks with coexisting chemical and physical cross-links of different structures have been attracting researchers interest [7,8,9,10]. To find the molecular distribution of cross-linked clusters (three-dimensional polymers), their average molecular weight, and the gel point, in terms of their system parameters such as concentration, temperature, reaction time, etc. in such multi-component polymer gels with complex cross-links, we have to know the relation between microscopic structure of the cross-link junctions and macroscopic change in thermodynamic state of the system. For a special case of chemical co-reaction between two species of polydisperse functional molecules, Stockmayer [11] derived, on the basis of the classical tree statistics, the molecular distribution and the weight-average molecular weight of the generated three-dimensional polymers in terms of the degree of reaction p,q for each functional group. From the result he presented the gel point condition for the formation of binary polymer networks. In this paper, we eliminate the assumption of pairwise reaction, and generalize his result to cross-links of arbitrary multiplicity in order to apply the theoretical model to real physical gels. Here the multiplicity is defined by a set of numbers of functional groups contained in a cross-link junction.

One of the important examples of such mixed multiple junctions is ion-binding cross-linking in which variable number of ligands attached on the polymer chains form ion complexes [12] around an added metal ion as a nucleus. Ions attract functional groups (ligands) on the polymer chains to form coordination complexes, which serve as network junctions [12,13]. When the concentration of added ions reaches a critical value, polymer solutions turn into gels. Typical examples of such ionically cross-linked gels are metalic cations (Ca${}^{2+}$, Cu${}^{2+}$, Zn${}^{2+}$, Fe${}^{3+}$, ⋯) and basic anions (B${}^{-}$(OH)${}_{4}$, Sb${}^{-}$(OH)${}_{6}$, ⋯) complexed by synthetic polymers (poly(vinyl alcohol), polyacrylamide, poly(acrylic acid), ⋯) [14,15,16,17,18,19], or natural polymers (polysaccharides, proteins, ⋯) [20,21,22,23,24,25] bearing functional groups. The characteristic properties, such as reversible gelation, phase separation, viscoelasticity, concentration fluctuations, etc. of these aqueous polymer solutions, originate in the detailed structure of the coordination complexes and variation of their stepwise equilibrium constants. For instance, gels are often observed to turn back to sols under excess presence of ions (reentrant sol-gel-sol transition) [26]. Similar reentrant transitions were also observed in polymer solutions with biomolecularly recognizable cross-link junctions [27].

Other important examples are amphiphilic co-networks [6] such as seen in aqueous solutions of hydrophobic polymers mixed with low-molecular weight surfactants. Interaction between polymers and surfactants has been a subject of great interest [28,29]. The problem was initially laid in studies of proteins associated with natural lipids and with synthetic surfactants. More recently, interaction of water-soluble synthetic polymers such as poly(ethylene oxide) (PEO) with ionic and non-ionic surfactants [30,31,32,33] have attracted interest of researchers because of scientific and technological implications. When polymers carry small fraction of hydrophobic groups, profound influence of added surfactants on the rheological properties has been reported. For example, the plateau modulus of hydrophobically modified PEO solutions exhibits a peak when sodium dodecyl sulfate (SDS) is added to solutions of low polymer concentrations [34]. It was aslo reported that the viscosity of hydrophobically modified methylcellulose solutions mixed with SDS exhibits similar peaks at low polymer concentrations [35,36]. These non-monotonic rheologies suggest that there is an optimal concentration for the surfactant binding depending on the miscibility of different species of hydrophobes within the same cross-link junction.

In view of these important multi-component gelling polymer solutions, general equation for finding the gel point in terms of their compositions and temperature is desirable. Historically, the gel point conditions of single component trifunctional molecules, and mixture of trifunctional and bifunctional molecules, were studied by Flory [37,38,39,40]. It was later generarized by Stockmayer to reactive molecules of arbitrary functionality for monodisperse [41] and polydisperse [42] cases. These studies (referred to as FS) were based on the fundamental assumptions such that (i) cross-link reactions between functional groups occur pairwisely by chemical (covalent) bonds with equal reactivity, and (ii) intramolecular reactions leading to cyclic structures are prevented in the pre-gel regime. FS was later extended by Fukui and Yamabe [43] to include multiple reactions of the one component functional groups in which cross-link junctions contain arbitrary number of functional groups. Gelation with multiple cross-linking was later treated for thermoreversible gels with physical cross-link junctions by Tanaka and Stockmayer [44,45] (referred to as TS).

On the other hand, gel point condition and molecular weight distribution in the systems of binary multifunctional molecules with co-polymerization between different functional groups were first studied by Stockmayer [11] for the mixtures of polyfunctional A and B molecules. Systematic methods for studying their average properties were later developed by Macosko and Miller [46]. Thermoreversible gelation with binary pairwise cross-links was studied by Tanaka et al. [45,47,48,49,50], and later extended to thermoreversible gels with multiple cross-link junctions [51,52].

It is well known that the decrease in the mixing entropy with growing tree-dimensional molecules drives the solution into a tendency to liquid–liquid phase separation. The relationship between gelation and phase separation has been relatively well understood [1,2,45], but progress continues to be made with some open interesting questions.

In this paper we describe detailed models and their applications to thermoreversible gelation of binary reactive molecules in a common solvent. We present equations for the gel point condition and the phase transition boundaries (spinodal lines) in terms of the microscopic parameters related to the structure of network junctions. Hence we can analyze the structure and strength of the network junctions from macroscopic measurements of phase transitions.

## 2. Stoichiometric Definition of the Model Solution

The model polymer solution we study here consists of two species of reactive molecules, referred to as R{A${}_{f}$}(A molecule) and R{B${}_{g}\}$ (B molecule), in a common solvent, mostly water. Molecules can be any type, such as high molecular weight linear polymers, star polymers, or low molecular weight polyfunctional molecules (cross-linkers), etc. An A molecule carries the number *f* of functional groups A (A groups), and a B molecule carries the number *g* of groups B (B groups). We can assume g≤f without losing generality, and refer to R{A${}_{f}$} as primary component, R{B${}_{g}$} as secondary one of the ternary solution. Functional groups are capable of forming complexes (mixed cross-link junctions) with binding numbers controlled by the thermodynamic conditions with equilibrium constants. Specific examples are polymers bearing hydrophobic short chains with different species or different sizes at their chain ends (mixtures of telechelic associating polymers). Hydrophobes form mixed micelles at cross-link junctions. Another examples are star polymers bearing functional groups (ligands) at their arm ends forming complexes with metal cations, etc. In the present study, solvent is assumed to be inactive.

Let $n_{A}$ be the number of statistical repeat units on an A molecule, and $n_{B}$ on a B molecule. The molecular weights of them are then $M_{A}=M_{0}^{\left(A\right)}n_{A}$ and $M_{B}=M_{0}^{\left(B\right)}n_{B}$, where $M_{0}^{\left(A\right)}$ and $M_{0}^{\left(B\right)}$ are the molecular weights of their statistical repeat units.

To treat concentration, we take the volume of a solvent molecule as the unit of volume, and assume for simplicity that the volume of the statistical repeat units on a functional molecule is the same as that of the solvent molecule, as conventionally assumed in a simple polymer solution theory. The total volume of the solution is then given by $\Omega =n_{A}N_{A}+n_{B}N_{B}+N_{0}$. Here, $N_{\alpha }$ is the number of molecules of the type α in the solution. The volume fraction of each component is $\phi_{A}=n_{A}N_{A}/\Omega$ for R{A${}_{f}$}, $\phi_{B}=n_{B}N_{B}/\Omega$ for R{B${}_{g}$}, and $\phi_{0}=N_{0}/\Omega$ for the solvent. The number concentration of A groups and B groups are then given by $\psi_{A}=f\phi_{A}/n_{A}$ and $\psi_{B}=g\phi_{B}/n_{B}$.

## 3. Equilibrium Condition for the Formation of Cross-Link Junctions

Let us consider formation of the network junctions J$\left(k_{1},k_{2}\right)$ containing the number $k_{1}$ of A groups and $k_{2}$ of B groups in the reversible chemical reaction (see Figure 1)
(1)$$k_{1}J\left(1,0\right)+k_{2}J\left(0,1\right)⇄J\left(k_{1},k_{2}\right)$$

Let $p_{k_{1},k_{2}}$ be the probability for an arbitrarily chosen A group to belong to a J$\left(k_{1},k_{2}\right)$ junction, and let $q_{k_{1},k_{2}}$ be that for a B group. They are the counterparts of the conventional reactivity of the functional groups. Obviously, trivial equality
(2)$$\psi_{A}p_{k_{1},k_{2}}/k_{1}=\psi_{B}q_{k_{1},k_{2}}/k_{2}$$
holds for the number of junctions specifid by $\left(k_{1},k_{2}\right)$.

For chemical cross-links, the reactivity obeys kinetic equation of the reaction, and hence $p_{k_{1},k_{2}}$ is a function of the reaction time. For physical cross-links, we may impose the equilibrium condition for their formation, and in an ideal case the reactivity must fulfil the equation
(3)$$\frac{\psi_{A}p_{k_{1},k_{2}}/k_{1}}{{\left(\psi_{A}p_{1,0}\right)}^{k_{1}}{\left(\psi_{B}q_{0,1}\right)}^{k_{2}}}=K_{k_{1},k_{2}}$$
for A groups in the reaction (Equation 1), where $K_{k_{1},k_{2}}$ is the equilibrium constant of the junctions of type $\left(k_{1},k_{2}\right)$. Hence we have
(4)$$\psi_{A}p_{k_{1},k_{2}}=k_{1}K_{k_{1},k_{2}}{\left(\psi_{A}p_{1,0}\right)}^{k_{1}}{\left(\psi_{B}q_{0,1}\right)}^{k_{2}}$$

Therefore, the reaction probability is given by
(5)$$p_{k_{1},k_{2}}=p_{1,0}k_{1}K_{k_{1},k_{2}}{z_{A}}^{k_{1}-1}{z_{B}}^{k_{2}}$$
where
(6)$$z_{A}\equiv \psi_{A}p_{1,0}\;z_{B}\equiv \psi_{B}q_{0,1}$$
are the concentration of the free functional groups that remain unreacted in the solution. Summing up for all possible $k_{1}\geq 1$ and $k_{2}\geq 0$, and using the normalization condition for the probability $p_{k_{1},k_{2}}$, we find
(7)$$\psi_{A}=z_{A}u_{A}\left(z_{A},z_{B}\right)$$
where the function defined by
(8)$$u_{A}\left(z_{A},z_{B}\right)\equiv \sum_{k_{1}\geq 1,k_{2}\geq 0}k_{1}K_{k_{1},k_{2}}{z_{A}}^{k_{1}-1}{z_{B}}^{k_{2}}$$
characterizes the microscopic structure of a junction containing at least one A group.

Similarly, from the identity (Equation 2), we find
(9)$$q_{k_{1},k_{2}}=q_{0,1}k_{2}K_{k_{1},k_{2}}{z_{A}}^{k_{1}}{z_{B}}^{k_{2}-1}$$
and hence
(10)$$\psi_{B}=z_{B}u_{B}\left(z_{A},z_{B}\right)$$
for B groups with
(11)$$u_{B}\left(z_{A},z_{B}\right)\equiv \sum_{k_{1}\geq 0,k_{2}\geq 1}k_{2}K_{k_{1},k_{2}}{z_{A}}^{k_{1}}{z_{B}}^{k_{2}-1}$$

The structure functions $u_{A},u_{B}$ have physical meanings of the reciprocal unreactivity $u_{A}\left(z_{A},z_{B}\right)=1/p_{1,0},u_{B}\left(z_{A},z_{B}\right)=1/q_{0,1}$.

Coupled Equations (Equation 7) and (Equation 10) show that the total numbers of A groups and B groups do not change in the process of cross-link formation, but only their connectivity is changed. Therefore in what follows we call them materials conservation law. To study solution properties, these coupled equations must be solved for the two unknown variables $z_{A},z_{B}$ as functions of the concentration $\psi_{A},\psi_{B}$ given in the preparation stage of the experiments.

Let us next introduce the reactivity *p* and *q* of the functional groups of each species by the definitions $p\equiv 1-p_{1,0}$ and $q\equiv 1-q_{0,1}$. They have the meaning of conventional reactivities, but extended to multiple reactions, such that for an arbitralily chosen functional group to be in any kind of cross-links. Obviously we have the relation
(12a)$$p=\sum_{k_{1}\geq 1,k_{2}\geq 1}p_{k_{1},k_{2}}$$
(12b)$$q=\sum_{k_{1}\geq 1,k_{2}\geq 1}q_{k_{1},k_{2}}$$

In terms of these reactivities, we can write the materials conservation law as
(13a)$$p=\sum_{k_{1}\geq 1,k_{2}\geq 1}\left(k_{1}K_{k_{1},k_{2}}\right){\psi_{A}}^{k_{1}^{\prime }}{\psi_{B}}^{k_{2}}{\left(1-p\right)}^{k_{1}}{\left(1-q\right)}^{k_{2}}$$
(13b)$$q=\sum_{k_{1}\geq 1,k_{2}\geq 1}\left(k_{2}K_{k_{1},k_{2}}\right){\psi_{A}}^{k_{1}}{\psi_{B}}^{k_{2}^{\prime }}{\left(1-p\right)}^{k_{1}}{\left(1-q\right)}^{k_{2}}$$
where abbreviated notations $k_{1}^{\prime }\equiv k_{1}-1$ and $k_{2}^{\prime }\equiv k_{2}-1$ have been introduced for simplicity.

## 4. The Average Branching Number and Response to the Concentration Fluctuations

Let us consider the average structure of cross-link junctions. Using Equations (Equation 5) and (Equation 9) for the reactivities $p_{k_{1},k_{2}},q_{k_{1},k_{2}}$, the average number of A groups in the cross-link junctions is calculated as
(14)$$\begin{array}{cc} {\bar{\mu }}_{A,A} & \equiv \sum_{k_{1},k_{2}}k_{1}p_{k_{1},k_{2}}=p_{1,0}\sum_{k_{1},k_{2}}{k_{1}}^{2}K_{k_{1},k_{2}}{z_{A}}^{k_{1}^{\prime }}{z_{B}}^{k_{2}}\\ & =p_{1,0}\left\{1+z_{A}\frac{\partial }{\partial z_{A}}\right\}u_{A}\left(z_{A},z_{B}\right)=1+\kappa_{A,A}\left(z_{A},z_{B}\right) \end{array}$$
where the element of $\hat{\kappa }$-matrix is defined by the logarithmic derivative
(15)$$\kappa_{\alpha ,\beta }\equiv \frac{\partial \ln u_{\alpha }}{\partial \ln z_{\beta }}$$
The relation $p_{1,0}u_{A}\left(z_{A},z_{B}\right)=1$ has been used.

Similarly, we have
(16)$${\bar{\mu }}_{B,B}\equiv \sum_{k_{1},k_{2}}k_{2}q_{k_{1},k_{2}}=1+\kappa_{B,B}\left(z_{A},z_{B}\right)$$
for the average number of B groups in the junctions. Other elements of the $\hat{\mu }$-matrix are defined by
(17b)$${\bar{\mu }}_{A,B}\equiv \sum_{k_{1},k_{2}}k_{2}p_{k_{1},k_{2}}=\kappa_{A,B}\left(z_{A},z_{B}\right)$$
(17b)$${\bar{\mu }}_{B,A}\equiv \sum_{k_{1},k_{2}}k_{1}q_{k_{1},k_{2}}=\kappa_{B,A}\left(z_{A},z_{B}\right)$$
although these do not have direct geometrical implication of the junction structure. The determinant of the $\hat{\mu }$-matrix
(18)$$D_{\mu }\equiv \left(1+\kappa_{A,A}\right)\left(1+\kappa_{B,B}\right)-\kappa_{A,B}\kappa_{B,A}$$
is related to the determinant of $\hat{\kappa }$-matrix
(19)$$D_{\kappa }\equiv \kappa_{A,A}\kappa_{B,B}-\kappa_{A,B}\kappa_{B,A}$$
by the equation
(20)$$D_{\mu }=1+\kappa_{A,A}+\kappa_{B,B}+D_{\kappa }$$

Let us next consider infinitesimal variation of the concentrations of unreacted groups in response to the variations $d\psi_{A},d\psi_{B}$ of the total concentrations. This depends on the average structure of the cross-link junction, and can be found by taking logarithmic derivatives of the materials conservation law (Equation 7) and (Equation 10). We have
(21)$$\left[\begin{array}{c} d\ln\psi_{A}\\ d\ln\psi_{B} \end{array}\right]=\left[\begin{array}{cc} 1+\kappa_{A,A} & \kappa_{A,B}\\ \kappa_{B,A} & 1+\kappa_{B,B} \end{array}\right]\left[\begin{array}{c} d\ln z_{A}\\ d\ln z_{B} \end{array}\right]$$

Solving for $d\ln z_{A}$ and $d\ln z_{B}$, we find
(22)$$\left[\begin{array}{c} d\ln z_{A}\\ d\ln z_{B} \end{array}\right]=\frac{1}{D_{\mu }}\left[\begin{array}{cc} 1+\kappa_{B,B} & -\kappa_{A,B}\\ -\kappa_{B,A} & 1+\kappa_{A,A} \end{array}\right]\left[\begin{array}{c} d\ln\psi_{A}\\ d\ln\psi_{B} \end{array}\right]$$

## 5. The Weight Average Molecular Weight of the Cross-Linked Clusters and the Gel Point Condition

To study solution properties, in particular gelation and phase separation, let us consider the distribution of cross-linked clusters (three-dimensional polymers) in the solution. Let $N_{l,m}$ be the number of clusters made up of the number *l* of R{A${}_{f}$} (A molecules) and *m* of R{B${}_{g}$} (B molecules). Their number concentration is given by
(23)$$\nu_{l,m}\equiv N_{l,m}/\Omega$$

The number-average molecular weight of the clusters is defined by
(24)$${\bar{M}}_{n}\equiv \sum_{l,m}\left(M_{A}l+M_{B}m\right)N_{l,m}/\sum_{l,m}N_{l,m}$$
and similarly the weight-average molecular weight is by
(25)$${\bar{M}}_{w}\equiv \sum_{l,m}{\left(M_{A}l+M_{B}m\right)}^{2}N_{l,m}/\sum_{l,m}\left(M_{A}l+M_{B}m\right)N_{l,m}$$

In our previous paper [51,52], we derived both averages for the mixtures capable of forming any types of multiple cross-links on the basis of classical tree statistics [11,37,38,39,40,41,42,43]. The assumption of pairwise reaction is eliminated to include multiple cross-links made up of any numbers of A and B groups. Under the simplifying assumption that the molecular weight $M_{0}^{\left(A\right)}$ of a statistical repeat unit of A molecules and $M_{0}^{\left(B\right)}$ of B molecules are the same ($\equiv M_{0}$), the result (Equation (3.22) in the literature [52]) is
(26)$$\begin{array}{ccc} \phi \frac{{\bar{M}}_{w}}{M_{0}}=n_{A}\phi_{A}+n_{B}\phi_{B} & + & \frac{1}{D}\left\{{n_{A}}^{2}\psi_{A}\left[\kappa_{A,A}-\left(g-1\right)D_{\kappa }\right]+{n_{B}}^{2}\psi_{B}\left[\kappa_{B,B}-\left(f-1\right)D_{\kappa }\right]\right\}\\ & + & \frac{n_{A}n_{B}}{D}\left(\psi_{A}\kappa_{A,B}+\psi_{B}\kappa_{B,A}\right) \end{array}$$
where $\phi \equiv \phi_{A}+\phi_{B}$ is the total solute volume fraction, $\kappa_{\alpha ,\beta }$ is the $\hat{\kappa }$-matrix defined above, and $D_{\kappa }$ is its determinant. The denominator in ${\bar{M}}_{w}$ is defined by
(27)$$D\left(z_{A},z_{B}\right)\equiv 1-f^{\prime }\kappa_{A,A}-g^{\prime }\kappa_{B,B}+f^{\prime }g^{\prime }D_{\kappa }$$
(different from $D_{\kappa }$, $D_{\mu }$). This function corresponds to the determinant of the matrix introduced by Gordon [53] in his cascade theory of gelation for mixtures of multi-component reactive molecules. Abbreviated notations $f^{\prime }\equiv f-1$ and $g^{\prime }\equiv g-1$ have been used since they will frequently appear in the following. This result is a generalization of the old Stockmayer’s average molecular weight [11] for the pairwise reaction to multiple ones. At the gel point, the weight average molecular weight goes to infinity, and hence we have
(28)$$D(z_{A},z_{B})=0$$
for a gel to appear (gel point condition). At the point satisfying this gel point condition, the solution changes from a sol state to a gel state, and vice versa. Such a reversible phase transition is called thermoreversible gelation, or sol-gel transition (referred to as SGT). Materials conservation law (Equation 7) and (Equation 10), together with the gel point condition (Equation 28), leads to the relation between $\psi_{A}$ and $\psi_{B}$, and therefore gives the SGT line on the ternary phase plane when parameters $z_{A}$ and $z_{B}$ are eliminated.

## 6. The Average Number of Molecules in the Cross-Linked Clusters and the Spinodal Condition for the Solution

### 6.1. Chemical Potentials Described in Terms of the Number-Average Degree of Polymerization

To study thermodynamic properties (in particular phase separation) of a polymer solution, we have to derive the chemical potentials of all components in the solution. In our present ternary reactive solutions, we can apply our general theoretical framework [45] for the sudy of associating polymer solutions. We omit its details here by decribing only the outline of derivation.

We start from the free energy of the model solution. It consists of the sum of the two parts
(29)$$\Delta F=\Delta_{\mathrm{cross}}F+\Delta_{\mathrm{mix}}F$$

The first term (per unit volume) is the free energy of cluster formation by cross-linking
(30)$$\Delta_{\mathrm{cross}}F/\Omega =\sum_{l,m}\Delta A_{l,m}\nu_{l,m}$$
where $\Delta A_{l,m}$ is the free energy to form an (l,m) cluster from the number *l* of A molecules and *m* of B molecules in the separated free state.

The second term is the free energy change
(31)$$\beta \Delta_{\mathrm{mix}}F/\Omega =\sum_{l,m}\nu_{l,m}\ln\phi_{l,m}+\phi_{0}\ln\phi_{0}+g\left(\left\{\phi \right\}\right)$$
upon mixing the formed clusters with the solvent. In this equation, $\phi_{l,m}\equiv \left(n_{A}l+n_{B}m\right)\nu_{l,m}$ is the volume fraction of the clusters, and $\beta \equiv 1/k_{B}T$ is the inverse temperature. The interaction term
(32)$$g\left(\left\{\phi \right\}\right)\equiv \left(\chi_{A,0}\phi_{A}+\chi_{B,0}\phi_{B}\right)\phi_{0}+\chi_{A,B}\phi_{A}\phi_{B}$$
gives enthalpy change due to the contact between different species of molecules. The interaction parameters $\chi_{\alpha ,\beta }\left(T\right)$ are Flory’s χ-parameters in the conventional polymer solution theory [40].

By the standard thermodynamic procedure of taking partial derivatives of the free energy, we find the chemical potentials $\Delta \mu_{l,m}$ for the clusters of (l,m) type. We then impose equilibrium condition
(33)$$\Delta \mu_{l,m}=l\Delta \mu_{1,0}+m\Delta \mu_{0,1}$$
for the cluster formation, and find the overall independent values $\Delta \mu_{A}\equiv \Delta \mu_{1,0}$ and $\Delta \mu_{B}\equiv \Delta \mu_{0,1}$ for each species. Together with the chemical potential for the solvent, we have
(34a)$$\frac{\beta \Delta \mu_{A}}{n_{A}}=\frac{1+\ln x}{n_{A}}-\nu^{S}+g_{A}\left(\left\{\phi \right\}\right)$$
(34b)$$\frac{\beta \Delta \mu_{B}}{n_{B}}=\frac{1+\ln y}{n_{B}}-\nu^{S}+g_{B}\left(\left\{\phi \right\}\right)$$
(34c)$$\beta \Delta \mu_{0}=1+\ln\phi_{0}-\nu^{S}+g_{0}\left(\left\{\phi \right\}\right)$$

Here, $x\equiv f\nu_{1,0}$ and $y\equiv g\nu_{0,1}$ are the concentration of the functional groups carried by the unassociated free molecules of each species, and
(35)$$\nu^{S}=\nu +\phi_{0}$$
is the total translational degree of freedom of solute and solvent molecules, where
(36)$$\nu \equiv \sum_{l,m}\nu_{l,m}$$
is the total number of connected clusters in a unit volume of the solution. The sum on the right hand side includes unrected molecules (1,0) and (0,1). The gel part is excluded because the upper limit goes to infinity for a gel.

The interaction terms are explicitly given by
(37a)$$g_{A}\left(\left\{\phi \right\}\right)=\chi_{A,0}\phi_{0}\left(1-\phi_{A,0}\right)-\chi_{B,0}\phi_{0}\phi_{B}+\chi_{A,B}\phi_{B}\left(1-\phi_{A}\right)$$
(37b)$$g_{B}\left(\left\{\phi \right\}\right)=\chi_{B,0}\phi_{0}\left(1-\phi_{B,0}\right)-\chi_{A,0}\phi_{0}\phi_{A}+\chi_{A,B}\phi_{A}\left(1-\phi_{B}\right)$$
(37c)$$g_{0}\left(\left\{\phi \right\}\right)=\left(\chi_{A,0}\phi_{A}+\chi_{B,0}\phi_{B}\right)\left(1-\phi_{0}\right)-\chi_{A,B}\phi_{A}\phi_{B}$$

The number concentrations x,y of unassociated free molecules can be related to the number concentration $z_{A},z_{B}$ of unassocated functional groups through the equations
(38a)$$x\equiv f\nu_{1,0}=\psi_{A}{p_{1,0}}^{f}=\frac{\psi_{A}}{u_{A}{\left(z_{A},z_{B}\right)}^{f}}=\frac{z_{A}}{u_{A}{\left(z_{A},z_{B}\right)}^{f^{\prime }}}$$
(38b)$$y\equiv g\nu_{0,1}=\psi_{B}{q_{0,1}}^{g}=\frac{\psi_{B}}{u_{B}{\left(z_{A},z_{B}\right)}^{g}}=\frac{z_{B}}{u_{B}{\left(z_{A},z_{B}\right)}^{g^{\prime }}}$$

Owing to the materials conservation law, x,y can be regarded as functions of the concentration {ϕ}.

### 6.2. Average Degree of Polymerization and Number Concentration of the Cross-Linked Clusters

Let us next consider the total degree ν of translational motion of clusters, i.e., the concentration of their centers of mass. Since a cluster of the type (l,m) includes the number *l* of A molecules and *m* of B molecules, the identity
(39)$$\sum_{l,m}\left(l+m\right)\nu_{l,m}=\frac{\phi_{A}}{n_{A}}+\frac{\phi_{B}}{n_{B}}$$
holds for the total number of molecules in the pre-gel regime. Hence we have
(40)$$\nu =\left(\frac{\phi_{A}}{n_{A}}+\frac{\phi_{B}}{n_{B}}\right){{\bar{P}}_{n}}^{-1}$$
where
(41)$${\bar{P}}_{n}\equiv \sum_{l,m}\left(l+m\right)\nu_{l,m}/\sum_{l,m}\nu_{l,m}$$
is the number-average degree of polymerization of cross-linked clusters. Because the number-average molecular weight is defined by Equation (Equation 24), ${\bar{P}}_{n}$ can be found by formally substituting $M_{A}=M_{B}=1$ in the calculation of ${\bar{M}}_{n}$. We have reported the latter in our preceding paper [52]. It is
(42)$$\frac{1}{{\bar{M}}_{n}}=\frac{fm_{A}}{M_{A}}\left(\frac{1}{f}+\frac{1}{{\bar{\mu }}_{n}^{\left(A\right)}}-1\right)+\frac{gm_{B}}{M_{B}}\left(\frac{1}{g}+\frac{1}{{\bar{\mu }}_{n}^{\left(B\right)}}-1\right)$$
where $m_{A}\equiv M_{A}N_{A}/\left(M_{A}N_{A}+M_{B}N_{B}\right),m_{B}\equiv M_{B}N_{B}/\left(M_{A}N_{A}+M_{B}N_{B}\right)$ are the mass fractions, and
(43a)$$\frac{1}{{\bar{\mu }}_{n}^{\left(A\right)}}\equiv \sum_{k_{1}\geq 1,k_{2}\geq 0}\frac{p_{k_{1},k_{2}}}{k_{1}+k_{2}}$$
(43b)$$\frac{1}{{\bar{\mu }}_{n}^{\left(B\right)}}\equiv \sum_{k_{1}\geq 0,k_{2}\geq 1}\frac{q_{k_{1},k_{2}}}{k_{1}+k_{2}}$$
are the reciprocal number-average multiplicity of the cross-link junctions. Setting $M_{A}=M_{B}=1$ in these equations, we find
(44)$$\nu =\psi_{A}\left(\frac{1}{f}+\frac{1}{{\bar{\mu }}_{n}^{\left(A\right)}}-1\right)+\psi_{B}\left(\frac{1}{g}+\frac{1}{{\bar{\mu }}_{n}^{\left(B\right)}}-1\right)$$

### 6.3. Spinodal Condition

The Gibbs condition for a liquid–liquid phase equilibrium (binodal condition) requires a balance of these three chemical potentials in two distinct liquid phases with different concentrations. Because of the many difficulties in solving the coupled equations for binodals, we instead consider the phase boundary of stability limit (spinodal condition) by constructing the Gibbs matrix $\hat{G}$ from these chemical potentials. Inside of the spinodal line, the solution becomes unstable against infinitesimal concentration fluctuations. The spinodal condition is simply given by a single equation $|\hat{G}|=0$, and easy to solve. In general, a cloud-point curve observed in an experiment is expected to be identical to the binodal, and may be different from the spinodal, but usually they lie close to each other.

Due to the Gibbs–Dühem relation, two of the three chemical potentials are independent. We choose the solvent as the reference component, and consider A and B as independent components. To avoid cumbersome treatment involving $\nu^{S}$, we make subtraction and consider the difference measured from the reference chemical potential of the solvent. Thus, we have
(45a)$$\beta \Delta \mu_{A}^{\prime }\equiv \beta \left(\Delta \mu_{A}-\Delta \mu_{0}\right)=\frac{1}{n_{A}}\ln x-\ln\phi_{0}+g_{A}-g_{0}+\mathrm{const}.$$
(45b)$$\beta \Delta \mu_{B}^{\prime }\equiv \beta \left(\Delta \mu_{B}-\Delta \mu_{0}\right)=\frac{1}{n_{B}}\ln y-\ln\phi_{0}+g_{B}-g_{0}+\mathrm{const}.$$

The change of lnx and lny in response to the infinitesimal variation $d\psi_{A},d\psi_{B}$ is found by simple differentiation as
(46)$$\left[\begin{array}{c} d\ln x\\ d\ln y \end{array}\right]=\left[\begin{array}{cc} 1-f^{\prime }\kappa_{A,A} & -f^{\prime }\kappa_{A,B}\\ -g^{\prime }\kappa_{B,A} & 1-g^{\prime }\kappa_{B.B} \end{array}\right]\left[\begin{array}{c} d\ln z_{A}\\ d\ln z_{B} \end{array}\right]$$

By using the transition matrix (Equation 22) from *z* to ψ, we find
(47)$$\left[\begin{array}{c} d\ln x\\ d\ln y \end{array}\right]=\frac{1}{D_{\mu }}\left[\begin{array}{cc} 1-f^{\prime }\kappa_{A,A}+\kappa_{B,B}-f^{\prime }D_{\kappa } & -f^{\prime }\kappa_{A,B}\\ -g^{\prime }\kappa_{B,A} & 1-g^{\prime }\kappa_{B.B}+\kappa_{A,A}-g^{\prime }D_{\kappa } \end{array}\right]\left[\begin{array}{c} d\ln\psi_{A}\\ d\ln\psi_{B} \end{array}\right]$$
where $D_{\kappa }$ is the determinant (Equation 19) of $\hat{\kappa }$-matrix.

Substituting this relation into the chemical potentials, and together with the equations
(48a)$$d\left(g_{A}-g_{0}\right)=-2\chi_{A,0}d\phi_{A}-\Delta \chi d\phi_{B}$$
(48b)$$d\left(g_{B}-g_{0}\right)=-\Delta \chi d\phi_{A}-2\chi_{B,0}d\phi_{B}$$
for the mixing enthalpy, with $\Delta \chi \equiv \chi_{A,0}+\chi_{B,0}-\chi_{A,B}$, we finally find
(49)$$\left[\begin{array}{c} d(\beta \Delta \mu_{A}^{\prime })\\ d(\beta \Delta \mu_{B}^{\prime }) \end{array}\right]=\left[\begin{array}{cc} \frac{1+\kappa_{B,B}-f^{\prime }\left(\kappa_{A,A}+D_{\kappa }\right)}{n_{A}\phi_{A}D_{\mu }}+\frac{1}{\phi_{0}}-2\chi_{A,0} & -\frac{f^{\prime }\kappa_{A,B}}{n_{A}\phi_{B}D_{\mu }}+\frac{1}{\phi_{0}}-\Delta \chi \\ -\frac{g^{\prime }\kappa_{B,A}}{n_{B}\phi_{A}D_{\mu }}+\frac{1}{\phi_{0}}-\Delta \chi & \frac{1+\kappa_{A,A}-g^{\prime }\left(\kappa_{B,B}+D_{\kappa }\right)}{n_{B}\phi_{B}D_{\mu }}+\frac{1}{\phi_{0}}-2\chi_{B,0} \end{array}\right]\left[\begin{array}{c} d\phi_{A}\\ d\phi_{B} \end{array}\right]$$

The first matrix factor on the right hand side is the Gibbs matrix $\hat{G}$.

After a lengthy calculation of its determinant, we find the spinodal condition as
(50)$$\begin{array}{cc} |\hat{G}| & =\frac{K_{A}K_{B}-f^{\prime }g^{\prime }\kappa_{A,B}\kappa_{B,A}}{n_{A}n_{B}\phi_{A}\phi_{B}{D_{\mu }}^{2}}+\frac{1}{\phi_{0}D_{\mu }}\left\{\frac{K_{A}}{n_{A}\phi_{A}}+\frac{K_{B}}{n_{B}\phi_{B}}+\frac{f^{\prime }\kappa_{A,B}}{n_{A}\phi_{B}}+\frac{g^{\prime }\kappa_{B,A}}{n_{B}\phi_{A}}-2\chi_{A,B}D_{\mu }\right\}\\ & -\;\frac{2}{D_{\mu }}\left\{\chi_{B,0}\frac{K_{A}}{n_{A}\phi_{A}}+\chi_{A,0}\frac{K_{B}}{n_{B}\phi_{B}}+\frac{\Delta \chi }{2}\left(\frac{f^{\prime }\kappa_{A,B}}{n_{A}\phi_{B}}+\frac{g^{\prime }\kappa_{B,A}}{n_{B}\phi_{A}}\right)\right\}-\tilde{\chi }=0 \end{array}$$
where
(51a)$$K_{A}=\left(1+\kappa_{B,B}\right)\left(1-f^{\prime }\kappa_{A,A}\right)+f^{\prime }\kappa_{A,B}\kappa_{B,A}$$
(51b)$$K_{B}=\left(1+\kappa_{A,A}\right)\left(1-g^{\prime }\kappa_{B,B}\right)+g^{\prime }\kappa_{A,B}\kappa_{B,A}$$

These are functions of $z_{A}$ and $z_{B}$, and hence functions of reactivities *p* and *q*. In what follows we therefore write the spinodal condition as
(52)$$S\left(p,q\right)\equiv |\hat{G}\left(z_{A},z_{B}\right)|=0$$

The ternary effective interaction parameter in the last term of (Equation 50) is defined by
(53)$$\tilde{\chi }\equiv {\chi_{A,B}}^{2}+{\chi_{A,0}}^{2}+{\chi_{B,0}}^{2}-2\chi_{A,B}\chi_{A,0}-2\chi_{A,B}\chi_{B,0}-2\chi_{A,0}\chi_{B,0}$$

From the materials conservation law, we can find *p* and *q* as functions of the volume fraction $\phi_{A}$ and $\phi_{B}$. Then the spinodal lines can be drawn on the ternary phase plane.

We can easily confirm that (Equation 50) reduces to the conventional Flory–Huggins equation (XIII-1d of [40]) for ternary spinodal condition
(54)$$S\left(p,q\right)=\frac{1}{n_{A}n_{B}\phi_{A}\phi_{B}}+\frac{1}{n_{A}n_{0}\phi_{A}\phi_{0}}+\frac{1}{n_{B}n_{0}\phi_{B}\phi_{0}}-2\left(\frac{\chi_{B,0}}{n_{A}\phi_{A}}+\frac{\chi_{A,0}}{n_{B}\phi_{B}}+\frac{\chi_{A,B}}{n_{0}\phi_{0}}\right)-\tilde{\chi }=0$$
by setting all κ elements to be zero.

In the special case of no solvent, we take the limit $\phi_{0}\to 0$ in (Equation 50), and find
(55)$$\frac{K_{A}^{\prime }}{n_{A}\phi_{A}}+\frac{K_{B}^{\prime }}{n_{B}\phi_{B}}-2\chi_{A,B}=0$$
with
(56a)$$K_{A}^{\prime }\equiv \left(K_{A}+g^{\prime }\frac{n_{A}}{n_{B}}\kappa_{B,A}\right)/D_{\mu }$$
(56b)$$K_{B}^{\prime }\equiv \left(K_{B}+f^{\prime }\frac{n_{B}}{n_{A}}\kappa_{A,B}\right)/D_{\mu }$$

This is the standard form of spinodal condition for binary associating solutions we have studied over the past decades [45].

### 6.4. Treatment of the Temperature

Although exploring the effect of temperature is not the main purpose of this study, we briefly summarize how to treat temperature within the present theoretical framework. Temperature comes in through the conventional interaction parameters $\chi_{\alpha ,\beta }$ and the new parameters $K_{k_{1},k_{2}}$ (equilibrium constants for cross-link formation). We regard A molecules as the primary component and B molecules as the secondary component because the latter is often an additional component mixed into the solution of the primary one. Therefore, we assume the conventional Shultz–Flory form
(57)$$\chi_{A,0}\left(T\right)=1/2-\psi \left(1-\Theta /T\right)$$
for the interaction between A molecules and solvent. Here, ψ is a material parameter of order unity, and Θ is the theta temperature of the primary solution. Because Θ does not include the effect of cross-linking, it is referred to as unperturbed theta temperature. The equilibrium constants can be in principle described in terms of the binding constants
(58)$$\lambda \left(T\right)=\lambda_{0}\exp\left(-\epsilon_{A}/k_{B}T\right),\;\mu \left(T\right)\equiv \mu_{0}\exp\left(-\epsilon_{B}/k_{B}T\right)$$
of functional group of each species in the cross-link junctions (expamples given below), where $\lambda_{0},\mu_{0}$ are the entropy contribution and $\epsilon_{A},\epsilon_{B}$ are the enthalpy of binding.

## 7. Fixed Multiplicity Model

Let us first consider the simplest model junction whose numbers $k_{1}$ and $k_{2}$ are fixed (fixed multiplicity model). Functional groups in the solution are therefore either in an unreacted state (1,0), (0,1), or belong to a junction of $\left(k_{1},k_{2}\right)$ type. For special case of pairwise reaction $\left(k_{1},k_{2}\right)=\left(1,1\right)$, this model reduces to the conventional chemical (covalent) bonds for which molecular distribution function of the cross-linked polymers and the gel point condition were studied by Stockmayer [11].

For the fixed multiplicity model, the number of cross-link junctions formed in the solution is $\psi_{A}p/k_{1}$ as seen from A groups, and $\psi_{B}q/k_{2}$ as seen from B groups. Since they are the same, we have a relation
(59)$$q=\frac{r}{R}p$$
where
(60)$$R\equiv \psi_{B}/\psi_{A}$$
is the relative concentration of the number of B groups per an A group, and
(61)$$r\equiv k_{2}/k_{1}$$
is the multiplicity ratio. The relative concentration *R* is often used for the gelation of polymers (A) induced by added agencies (B), such as metallic cations, basic anions, surfactants, and other cross-linkers. The stoichiometric concentration is given by the condition R=r, and hence at this concentration the reactivities of the functional groups take the same value p=q. In what follows we use the scaled relative concentration
(62)ξ≡R/r
in order to see the physical meanings of the concentrations easily. For the low concentration region 0<ξ≤1, the reactivities change in the range 0<p≤ξ,0<q≤1, while for the high concentration region 1<ξ, they change in the region 0<p≤1,0<q≤1/ξ. The stoichiometric concentration is given by ξ=1.

### 7.1. Sol-Gel Transition (SGT) and Spinodals for the Fixed Multiplicity Model

For the fixed multiplicity model, the materials conservation law takes the form
(63a)$$p=\left(k_{1}K\right){\psi_{A}}^{k_{1}^{\prime }}{\psi_{B}}^{k_{2}}{\left(1-p\right)}^{k_{1}}{\left(1-q\right)}^{k_{2}}$$
(63b)$$q=\left(k_{2}K\right){\psi_{A}}^{k_{1}}{\psi_{B}}^{k_{2}^{\prime }}{\left(1-p\right)}^{k_{1}}{\left(1-q\right)}^{k_{2}}$$
where only one equilibrium constant $K\equiv K_{k_{1},k_{2}}$ is necessary. Eliminating *q* by using the stoichiometric relation (Equation 59), we find the equation
(64)$$F\left(p\right)\equiv p-\left(k_{1}K\right){\psi_{A}}^{k}r^{k_{2}}{\left(1-p\right)}^{k_{1}}{\left(\xi -p\right)}^{k_{2}}=0$$
which should be solved for the unknown variable *p* as a function of the controlled parameters $\psi_{A}$ and ξ fixed in the preparation stage of the solution. The new multiplicity sum
(65)$$k\equiv k_{1}+k_{2}-1$$
has been introduced.

We next find the gel point condition. Simple calculation leads to
(66)$$\hat{\kappa }=\left[\begin{array}{cc} k_{1}^{\prime }p, & k_{2}\;p\\ k_{1}q, & k_{2}^{\prime }q \end{array}\right]$$
and hence
(67)$$D_{\kappa }=-kpq$$
and
(68)$$D_{\mu }=1+{k_{1}}^{\prime }p+{k_{2}}^{\prime }q-kpq$$

From these results we find
(69)$$D\left(p,q\right)=1-f^{\prime }k_{1}^{\prime }p-g^{\prime }k_{2}^{\prime }q-f^{\prime }g^{\prime }kpq=0$$
for the gel point.

The *K* functions for the study of spinodal condition are then found to be
(70a)$$K_{A}=1-f^{\prime }k_{1}^{\prime }p+k_{2}^{\prime }q+f^{\prime }kpq$$
(70b)$$K_{B}=1-g^{\prime }k_{2}^{\prime }q+k_{1}^{\prime }p+g^{\prime }kpq$$

The spinodal condition is also found in the form of S(p,q)=0 by using these specific forms.

Since the gel point condition (Equation 69) is a second order algebraic equations for *p*, the solution can be found easily as
(71)$$p=p_{c}\left(\xi \right)=\frac{f^{\prime }k_{1}^{\prime }\xi +g^{\prime }k_{2}^{\prime }}{2f^{\prime }g^{\prime }k}\left\{-1+\sqrt{1+\frac{4f^{\prime }g^{\prime }k\xi }{{\left(f^{\prime }k_{1}^{\prime }\xi +g^{\prime }k_{2}^{\prime }\right)}^{2}}}\right\}$$
and $q=q_{c}\left(\xi \right)=p_{c}\left(\xi \right)/\xi$. The concentration of A groups at the gel point is then given by
(72)$$\psi_{A,c}={\left\{\frac{p_{c}\left(\xi \right)}{\left(k_{1}K\right)r^{k_{2}}{\left(1-p_{c}\left(\xi \right)\right)}^{k_{1}}{\left(\xi -p_{c}\left(\xi \right)\right)}^{k_{2}}}\right\}}^{1/k}$$

This result of an SGT line on the $\left(\psi_{A},\xi \right)$ plane can be mapped onto the ternary phase plane $\left(\phi_{A},\phi_{B},\phi_{0}\right)$ of the volume fraction in the following two ways in accordance with the methods by which the solutions are prepared in the experiments.

The first method of preparation is mixing a solution of R{A${}_{f}$}/S and a solution of R{B${}_{g}$}/S with a common concentration (volume fraction) ϕ with the volume ratio 1:*u*. The result leads to
(73)$$\phi_{A}=\phi \left(1-u\right),\;\phi_{B}=\phi u,\;\phi_{0}=1-\phi$$
where $\phi \equiv \phi_{A}+\phi_{B}$ is the total solute volume fraction. We then have
(74)R=αu/(1−u),ξ=βu/(1−u)
where $\alpha \equiv gn_{A}/fn_{B}$ depends only on the functionality and molecular weight of the solute molecules, and $\beta \equiv \alpha /r=gn_{A}k_{2}/fn_{B}k_{1}$. (Do not confuse with $1/k_{B}T$.) Since ξ depends only on *u*, we find $\phi_{c}=\phi_{c}\left(u\right)$ from (Equation 72).

The second method of preparation is to add B molecules R{B${}_{g}$}, or its solution R{B${}_{g}$}/S of a known concentration, to a pure R{A${}_{f}$}/S solution, as in titration experiments. In particular for adding B molecules, we have
(75)$$\phi_{A}=\phi \left(1-u\right),\;\phi_{B}=u,\;\phi_{0}=\left(1-\phi \right)\left(1-u\right)$$
and hence
(76)R=αu/ϕ(1−u),ξ=βu/ϕ(1−u)

### 7.2. Optimal Gelation Concentration

We have developed a very general strategy to study the relationship between microscopic structure of the cross-link junctions and macroscopic SGT in ternary polymer solutions. As an example of its practical applications, we try to find a minimum concentration of the functional A molecules, mostly functional polymers, necessary for driving the solutions into gel when cross-linking agency is added. If a minimum exists, it provides the most efficient concentration to obtain gelling solution from a viewpoint of materials designing.

To study this problem of optimizing gelation, let us examine the gel point condition (Equation 69). For an infinitesimal change $d\psi_{A}$ and $d\psi_{B}$, the reactivities changes along SGT line under the condition
(77)$$-dD=f^{\prime }\left(g^{\prime }kq+k_{1}^{\prime }\right)pd\ln p+g^{\prime }\left(f^{\prime }kp+k_{2}^{\prime }\right)qd\ln q=0$$

On the other hand, from the materials conservation law (Equation 63) and ([Disp-formula FD1000-gels-07-00089]), we find
(78)$$\left[\begin{array}{cc} \frac{1+k_{1}^{\prime }p}{1-p}, & \frac{k_{2}q}{1-q}\\ \frac{k_{1}p}{1-p}, & \frac{1+k_{2}^{\prime }q}{1-q} \end{array}\right]\left[\begin{array}{c} d\ln p\\ d\ln q \end{array}\right]=\left[\begin{array}{cc} k_{1}^{\prime }, & k_{2}\\ k_{1}, & k_{2}^{\prime } \end{array}\right]\left[\begin{array}{c} d\ln\psi_{A}\\ d\ln\psi_{B} \end{array}\right]$$

Solving for dlnp and dlnq, we find
(79)$$\left[\begin{array}{c} d\ln p\\ d\ln q \end{array}\right]=\frac{\left(1-p)(1-q\right)}{1+k_{1}^{\prime }p+k_{2}^{\prime }q-kpq}\left[\begin{array}{cc} \frac{k_{1}^{\prime }-kq}{1-q}, & \frac{k_{2}}{1-q}\\ \frac{k_{1}}{1-p}, & \frac{k_{2}^{\prime }-kp}{1-p} \end{array}\right]\left[\begin{array}{c} d\ln\psi_{A}\\ d\ln\psi_{B} \end{array}\right]$$

Upon substitution into minimum SGT condition (Equation 77), we find
(80)$$H\left(p,q\right)\equiv H_{A}\left(p,q\right)d\ln\psi_{A}+H_{B}\left(p,q\right)d\ln\psi_{B}=0$$
where
(81a)$$H_{A}\left(p,q\right)\equiv f^{\prime }p\left(g^{\prime }kq+k_{1}^{\prime }\right)\frac{k_{1}^{\prime }-kq}{1-q}+g^{\prime }q\left(f^{\prime }kp+k_{2}^{\prime }\right)\frac{k_{1}}{1-p}$$
(81b)$$H_{B}\left(p,q\right)\equiv f^{\prime }p\left(g^{\prime }kq+k_{1}^{\prime }\right)\frac{k_{2}}{1-q}+g^{\prime }q\left(f^{\prime }kp+k_{2}^{\prime }\right)\frac{k_{1}^{\prime }-kp}{1-p}$$

To find the concentration of B molecules which minimizes the concentration of A molecules along the SGT line, we fix $d\ln\psi_{A}=0$ in this equation, and have
(82)$$H_{B}\left(p,q\right)=0$$

The solution $\left(p^{*},q^{*},\xi^{*}\right)$ of the coupled Equations (Equation 64), (Equation 69) and (Equation 82) gives the optimal SGT.

### 7.3. Specific Models of the Cross-Link Junctions with Fixed Multiplicity

#### 7.3.1. Pairwise Cross-Links

Let us start from the simplest case of pairwise reaction $\left(k_{1},k_{2}\right)=\left(1,1\right)$. We have r=1,k=1,ξ=R. The elements of $\hat{\kappa }$-matrix are $\kappa_{A,A}=\kappa_{B,B}=0,\kappa_{A,B}=p,\kappa_{B,A}=q$, and therefore lead to $D_{\kappa }=-pq$ and $K_{A}=1+f^{\prime }pq,K_{B}=1+g^{\prime }pq$. Obviously, the gel point condition reduces to the monodisperse case of Stockmayer’s result [11]
(83)$$D\left(p,q\right)=1-f^{\prime }g^{\prime }pq=0$$

The optimal condition (Equation 82) then gives
(84)$$p^{*}=\frac{f^{\prime }g^{\prime }+1}{2f^{\prime }g^{\prime }}$$

From two other equations we find
(85)$$\psi_{A}^{*}=\frac{4f^{\prime }g^{\prime }}{K{\left(f^{\prime }g^{\prime }-1\right)}^{2}},\;R^{*}=\frac{{\left(f^{\prime }g^{\prime }+1\right)}^{2}}{4f^{\prime }g^{\prime }}$$

Gel formation is limited to a certain concentration region between $R_{\min}$ and $R_{\max}$, where $R_{\min}=1/f^{\prime }g^{\prime }$ and $R_{\max}=f^{\prime }g^{\prime }$. It is natural that we need a minimum amount of B component (lower bound $R_{\min}$) to drive the solution into gel, but a certain condition is necessary to see that the gel turns back to sol for excess B component at high B concentration (called reentrant sol-gel-sol transition, referred to as SGST). Under excess B molecules, A molecules are covered by B molecules and separated from each other rather than forming connected clusters because reactions among B groups ((0,2) etc.) are prohibited (see Figure 2).

In the following study, we attempt to find, in terms of the multiplicity of the cross-links, the necessary condition for the appearance of an optimal SGT point followed by a reentrant sol phase. The optimal SGT is practically important because at this point the solution can turn into a gel with a minimum amount of A molecules.

The spinodal condition for the fixed multiplicity model is given by (Equation 50) with $K_{A},\;K_{B}$ of Equations (Equation 70) and ([Disp-formula FD1001-gels-07-00089]). We find that it takes the form
(86)$$\begin{array}{cc} & \frac{1+(f^{\prime }+g^{\prime })pq}{n_{A}n_{B}\phi_{A}\phi_{B}{\left(1-pq\right)}^{2}}+\frac{1}{\phi_{0}\left(1-pq\right)}\left[\frac{1+(f^{\prime }p+n_{A}g^{\prime }/n_{B})q}{n_{A}\phi_{A}}+\frac{1+(g^{\prime }q+n_{B}f^{\prime }/n_{A})p}{n_{B}\phi_{B}}-2\chi_{A,B}\left(1-pq\right)\right]\\ & -\frac{1}{1-pq}\left[2\chi_{A,0}\frac{1+f^{\prime }pq}{n_{A}\phi_{A}}+2\chi_{B,0}\frac{1+g^{\prime }pq}{n_{B}\phi_{B}}+\Delta \chi \left(\frac{f^{\prime }p}{n_{A}\phi_{B}}+\frac{g^{\prime }q}{n_{B}\phi_{A}}\right)\right]-\tilde{\chi }=0 \end{array}$$
for pairwise association.

In the particular case of no solvent, this reduces to
(87)$$\frac{1+(f^{\prime }p+n_{A}g^{\prime }/n_{B})q}{n_{A}\phi_{A}\left(1-pq\right)}+\frac{1+(g^{\prime }q+n_{B}f^{\prime }/n_{A})p}{n_{B}\phi_{B}\left(1-pq\right)}-2\chi_{A,B}=0$$

If we regard B molecules as monofunctional molecules such as hydrogen-bonding water, low molecular weight surfactant, etc., we can set $g^{\prime }=0$ in this equation, and find our previous results [45,54,55] for the spinodal
(88)$$\frac{1}{n_{A}\phi_{A}}+\frac{1+[f^{\prime }n_{B}/n_{A}+{\left(fn_{B}/n_{A}\right)}^{2}]p}{n_{B}\phi_{B}\left(1-pq\right)}-2\chi_{A,B}=0$$
with $q=(fn_{B}\phi_{A}/gn_{A}\phi_{B})p$.

Figure 3a,b show how SGST and spinodals change on the ternary phase plane with the association constant *K* and the interaction parameters χ. In contrast to the conventional triangular phase plane, we use throughout this paper two new independent concentration variables, the volume fraction of $\phi_{A}$ of the A molecules (vertical axis) and the concentration of the B functional groups $\xi \equiv R/r=\beta \phi_{B}/\phi_{A}$ relative to that of A functional groups (horizontal axis). The stoichiometric concentration of the functional groups is given by ξ=1. The origin of the coordinate corresponds to the pure solvent, the point (0,1) to pure A component, and (∞,0) to pure B component. The phase region is limited from above by the condition $\phi_{0}=1-\left(\phi_{A}+\phi_{B}\right)=0$ (thin line). The main reason for using this phase plane is that in the experiments B molecules are often the third components added to the already prepared solutions of A molecules. However, we can use it for any ternary solutions instead of the conventional triangular plane without losing generality.

In genaral, the association constant *K* changes with solvent quality, so that it depends on the interaction parameters. The change is mainly caused by the exclusion of solvent molecules from the junction zones upon forming cross-linking bonds, leading to the change in the number of molecular contact. In the present numerical study, however, we have fixed it for simplicity independently of the χ parameters. Further scrutiny on this point from physical viewpoint will provide a clue to clarify the dependence of thermoreversible gelation on the solvent quality.

Figure 3a shows how the spinodal lines shift with increase of $\chi_{A,B}$, the interaction parameter between A- and B molecules, while the association constant is fixed, for a symmetric case of f=g=3 and $n_{A}=n_{B}=5$. In this test calculation, we have fixed K=30 and changed $\chi_{A,B}$ from 2.0 to 5.0. The sol region (below the red solid line) and the gel region (above it) are separated by a nonmonotonic line with a minimum, so that the sol-gel transition is a reentrant SGST. The minimum point is specifically given by the coordinates (25/16,8/81) = (1.56,0.099) according to Equation (Equation 85). With increase in $\chi_{A,B}$ the phase-separated two-phase region (the region above the broken line) expands because repelsive force between the two components becomes stronger.

Figure 3b shows the same as (a), but *K* is changed from 30 to 100 while $\chi_{A,B}$ is fixed at 5.0. We can see how the gel region expands with *K*, with flatter bottom part. The spinodals also shift with *K* because the repulsive force is compensated by increasing association [49].

Figure 4 shows the same as Figure 3a but here the interaction parameters between components are different. For a solvent which is poor for A but good for B, we have fixed as $\chi_{A,0}=2.0,\chi_{A,B}=\chi_{B,0}=0$ (blue broken line), and for an opposite case, we have fixed as $\chi_{B,0}=2.0,\chi_{A,B}=\chi_{A,0}=0$ (green broken line). In the former case, the solution starts from a two-phase state, whose phase-saparated region shrinks with increase in the concentration of B molecules, and eventually merges into one phase. The latter case shows an opposite tendency.

The turning points of the spinodal curves in the phase diagram, and in all phase diagrams presented in the following, lie close to the crititical point of phase separation. However, they do not exactly correspond to the critical points because our ternary co-ordinates are different from the conventional triangular ones.

#### 7.3.2. Mononuclear A Cross-Links

In order to study the reentrant SGST in more detail, we next consider cross-links containing only one A functional group (functional group carried by the molecules of higher functionality). It is an example of mononuclear cross-links since it has only one A group (referred to as nucleus) in a junction. We have $r=k_{2},\;k_{1}^{\prime }=0,k=k_{2}$ and $K_{A}=1+\left(k_{2}^{\prime }+f^{\prime }k_{2}p\right)q,\;K_{B}=1-g^{\prime }\left(k_{2}^{\prime }-k_{2}p\right)q,\;D_{\mu }=1+\left(k_{2}^{\prime }-k_{2}p\right)q$. The gel point condition leads to
(89)$$p_{c}\left(\xi \right)=\frac{k_{2}^{\prime }}{2f^{\prime }k_{2}}\left\{-1+\sqrt{1+\frac{4f^{\prime }k_{2}\xi }{g^{\prime }{k_{2}^{\prime }}^{2}}}\right\}$$
for the critical reactivity. The SGST line on the concentration plane takes the form
(90)$$\psi_{A,c}\left(\xi \right)={\left\{\frac{p_{c}\left(\xi \right)}{\left(k_{2}K\right)\left(1-p_{c}\left(\xi \right)\right){\left(\xi -p_{c}\left(\xi \right)\right)}^{k_{2}}}\right\}}^{1/k_{2}}$$

The optimal SGST condition becomes
(91)$$H_{B}\left(p\right)=-2f^{\prime }g^{\prime }{\left(k_{2}p\right)}^{2}+\left[g^{\prime }\left(f^{\prime \prime }k_{2}^{\prime }+f^{\prime }k_{2}\right)+1\right]\left(k_{2}p\right)+k_{2}^{\prime }\left(g^{\prime }k_{2}^{\prime }-1\right)=0$$
where $f^{\prime \prime }\equiv f-2$, and hence
(92)$$k_{2}p^{*}=\frac{g^{\prime }\left(f^{\prime \prime }k_{2}^{\prime }+f^{\prime }k_{2}\right)+1}{2f^{\prime }g^{\prime }}\left\{1+\sqrt{1+\frac{4f^{\prime }g^{\prime }k_{2}^{\prime }\left(g^{\prime }k_{2}^{\prime }-1\right)}{{\left[g^{\prime }\left(f^{\prime \prime }k_{2}^{\prime }+f^{\prime }k_{2}\right)+1\right]}^{2}}}\right\}$$

The lower bound of the gel region can be found by putting $q_{c}\left(\xi \right)=1$. We find $\xi_{\min}=0$ except a special case of $k_{2}=1$ (pairwise cross-link studied above). Since the number concentration of A groups is limited in the region below the maximum value $Kf/n_{A}$ corresponding to the volume fraction $\phi_{A}=1$, $\xi_{\min}$ is practically limited by this finite bound. Gelation becomes reentrant one (see Figure 5. The reentrant part lies beyond the region shown in the graph). The upper bound of the gel region can be found by setting $p_{c}\left(\xi \right)=1$, from which condition we have
(93)$$\xi_{\max}=g^{\prime }\left(f^{\prime }k_{2}+k_{2}^{\prime }\right)$$

Figure 5a shows SGST line (red solid line) and spinodal lines for three cases of the interaction parameters as above (broken lines) for a fixed association constants of a symmetric case of f=g=3 and $n_{A}=n_{B}=5$. In this test calculation, we have fixed K=20 and changed the quality of the solvent. The sol region (below the red solid line) and the gel region (above it) are separated by a line with a minimum (lying beyond the region shown in the graph). Figure 5b shows the reactivity *p* and *q* as functions of the scaled B concentration for three fixed volume fractions of A molecules $\phi_{A}=0.05,0.10,0.15$. They all start from 0. The reactivity of B groups shows a maximum near the stoichiometric concentration.

#### 7.3.3. Mononuclear B Cross-Links

We next consider model cross-links containing only one B group (functional group carried by the molecules of lower functionality), referred to as mononuclear B cross-links. We have $k_{2}^{\prime }=0,k=k_{1}$ and
(94)$$K_{A}=1-f^{\prime }\left(k_{1}^{\prime }+k_{1}q\right)p,\;K_{B}=1+\left(k_{1}^{\prime }+g^{\prime }k_{1}q\right)p,\;D_{\mu }=1+\left(k_{1}^{\prime }-k_{1}q\right)p$$

The gel point condition becomes
(95)$$D\left(p\right)=\xi -f^{\prime }k_{1}^{\prime }\xi p-f^{\prime }g^{\prime }k_{1}p^{2}=0$$
so that the reactivity of A groups along the SGT line is given by
(96)$$p_{c}\left(\xi \right)=\frac{k_{1}^{\prime }\xi }{2g^{\prime }k_{1}}\left\{-1+\sqrt{1+\frac{4g^{\prime }k_{1}}{f^{\prime }{k_{1}^{\prime }}^{2}\xi }}\right\}$$

The concentration of A groups along this SGT is
(97)$$\psi_{A,c}\left(\xi \right)={\left\{\frac{p_{c}\left(\xi \right)}{K{\left(1-p_{c}\left(\xi \right)\right)}^{k_{1}}\left(\xi -p_{c}\left(\xi \right)\right)}\right\}}^{1/k_{1}}$$

Therefore, the gel region is limited from below by
(98)$$\xi_{\min}=1/f^{\prime }\left(g^{\prime }k_{1}+k_{1}^{\prime }\right)$$
but there is no upper bound except the special case of $k_{1}=1$ (pairwise cross-link studied above). The SGT decreases monotonically with the concentration of B groups (see Figure 6). Mixing B molecules monotonically promotes gelation.

For special case of $k_{1}=2$, the cross-link takes the -ABA- form. Let us call it sandwich cross-link. Poly(vinyl alcohol) and polysaccharides cross-linked by borax ions fall on this categoly. In fact, phase diagrams similar to this one were already reported for garactomannan/borax solutions [21]. There is a uniform clear gel region at low concentration of B molecules before the solution becomes turbid. However, there is a difference such that monocomplexes of the type -AB may coexist in such ion-binding cross-links [26] while in the present fixed multiplicity model they are excluded. SGT line may have a minimum outside of the borax concentration region measured in the experiments [21] although it is not evident in the reported phase diagram.

In Figure 6, incidentally $f,n_{A}$ and $g,n_{B}$ are the same for these examples. Therefore, if vertical coordinate $\phi_{A}$ is replaced by $\phi_{B}$ and horizontal coordinate $\xi =\beta \phi_{B}/\phi_{A}$ is replaced by $\xi =\beta \phi_{A}/\phi_{B}$, Figure 5 (mononuclear A) and Figure 6 (mononuclear B) give the same phase diagram. However, because the concentration region covered is completely different, Figure 5 shows A-solvent edge and Figure 6 shows B-solvent edge of the conventional triangular diagram.

#### 7.3.4. Monofunctional B Molecules

In some important gelling polymer solutions, the added cross-linking molecules are monofunctional molecules (g=1). Metal ions forming coordination complexes with polymers carrying various ligands such as acrylic groups, hydroxy groups etc. fall onto this category. Surfactant molecules mixed with hydrophobically-modified water-soluble polymers (associating polymers) are also regarded as a typical example of such gelling solutions, because a surfactant molecule usually carries only one hydrophobe. We therefore here summarize the nature of SGT for the special case of g=1. The functionality *f* and the junction multiplicity $k_{2}$ can be arbitrary, but $k_{1}$ is assumed to be $k_{1}\geq 2$ in order to have gels. We have
(99)$$K_{A}=1-f^{\prime }k_{1}^{\prime }p+k_{2}^{\prime }q+f^{\prime }kpq,\;K_{B}=1+k_{1}^{\prime }p,\;D_{\mu }=1+k_{1}^{\prime }p+k_{2}^{\prime }q-kpq$$

Because $g^{\prime }=0$, the gel point condition gives a constant reactivity
(100)$$p_{c}=1/f^{\prime }k_{1}^{\prime }$$
along the SGT line, and hence the concentration of A groups along it is
(101)$$\psi_{A,c}\left(\xi \right)=\frac{f^{\prime }k_{1}^{\prime }}{{\left\{\left(k_{1}K\right)r^{k_{2}}{\left(f^{\prime }k_{1}^{\prime }-1\right)}^{k_{1}}{\left(f^{\prime }k_{1}^{\prime }\xi -1\right)}^{k_{2}}\right\}}^{1/k}}$$

There is only a lower bound $\xi_{\min}=1/f^{\prime }k_{1}^{\prime }$. The SGT line monotonically decreases. The gelation becomes easier in proportion to the concentration of added cross-linkers (see Figure 7).

In our previous study on ion-binding polymer solutions [26], we showed the existence of an optimal gelation point where a polymer solution turns into gel at the minimum polymer concentration when metal ions, regarded as g=1, are added. The appearance of such an optimal gel point was attributed to the effect of monocomplexes, i.e., complexes including only one ligand. From the viewpoint of ions, a monocomplex is formed by the adsorption of an ion onto one of the ligands carried by polymer chains without forming a cross-link. In the present case of g=1, it corresponds to (1,1) cross-link junction. If (1,1) is allowed in addition to (2,1), most of A groups are covered by B groups at their high concentration region, so that connected clusters are separated into smaller ones. In the fixed multiplicity model studied in this section, however, formation of (1,1) junction is excluded because only one type of cross-links with fixed multiplicity $\left(k_{1},k_{2}\right)$ for $k_{1}\geq 2$ is allowed. In other words, if cross-link junctions of $\left(1,k_{2}\right)$ are allowed to coexist with those of $k_{1}\geq 2$, an optimal gel point may appear. Details of such a variable multiplicity problem will be studied in the following sections.

Figure 7 shows the phase diagram of polyfunctional molecules ($f=n_{A}=10$) cross-linked by monofunctional small B molecules ($n_{B}=1,g=1$) by a sandwich type (2,1) multiplicity. The interaction parameter $\chi_{A,B}$ is varied (blue broken line) in a good solvent. SGT is monotonic with the concentration of the cross-linkers. Clear gel appears first, and then phase separation takes place. The clear gel region shrinks with increase in $\chi_{A,B}$. This type of phase diagrams are also frequently observed in the experiments.

## 8. Variable Multiplicity Model

Let us next study gelation with simultaneous formation of cross-links of different type. The multiplicity index $\left(k_{1},k_{2}\right)$ are allowed to vary under the thermodynamic equilibrium condition determined by the intermolecular interactions among the functional groups. In what follow, we consider several important cases frequently observed in experiments.

### 8.1. Completely Immiscible Cross-Link Junctions

In this case, there is no association between functional groups of different species. Only cross-links of the pure type $\left(k_{1},0\right)$ and $\left(0,k_{2}\right)$ are allowed. The polymer solution turns into an interpenetrating polymer networks (IPNs) when concentrations of both A and B groups exceed the critical values. Let us assume that multiplicities $k_{1},k_{2}$ are variable. Then, the IPN considered here is an extension of the conventional chemical IPN to more general physical gels with multiple junctions.

We have
(102a)$$u_{A}\left(z_{A}\right)=\sum_{k_{1}\geq 1}k_{1}K_{k_{1},0}{z_{A}}^{k_{1}^{\prime }}$$
(102b)$$u_{B}\left(z_{B}\right)=\sum_{k_{2}\geq 1}k_{2}K_{0,k_{2}}{z_{B}}^{k_{2}^{\prime }}$$

Hence,
(103a)$$\kappa_{A,A}=\frac{\partial \ln u_{A}\left(z_{A}\right)}{\partial \ln z_{A}}$$
(103b)$$\kappa_{B,B}=\frac{\partial \ln u_{B}\left(z_{B}\right)}{\partial \ln z_{B}}$$
and $\kappa_{A,B}=\kappa_{B,A}=0$. For spinodal conditions, *K* functions are given by
(104a)$$K_{A}=\left(1+\kappa_{B,B}\right)\left(1-f^{\prime }\kappa_{A,A}\right)$$
(104b)$$K_{B}=\left(1+\kappa_{A,A}\right)\left(1-g^{\prime }\kappa_{B,B}\right)$$
and
(105)$$D_{\mu }=\left(1+\kappa_{A,A}\right)\left(1+\kappa_{B,B}\right)$$

The gel point condition is factorized as
(106)$$D\left(z_{A},z_{B}\right)=\left\{\psi_{A}-f^{\prime }{z_{A}}^{2}u_{A}^{\prime }\left(z_{A}\right)\right\}\left\{\psi_{B}-g^{\prime }{z_{B}}^{2}u_{B}^{\prime }\left(z_{B}\right)\right\}=0$$
where abbreviated notation $u^{\prime }\left(z\right)\equiv du\left(z\right)/dz$ has been used. Accordingly, the concentration plane is separated into four regions of SOL, A-GEL, B-GEL and IPN(A/B-GEL).

To find SGT of each component, let us introduce association constants λ(T) and μ(T) for A- and B groups as in our previous study on one component thermoreversible gels [44,47]. They depend on the temperature *T*. We then have
(107)$$K_{k_{1},0}=\gamma_{k_{1}}^{\left(A\right)}{\lambda (T)}^{k_{1}^{\prime }},\;K_{0,k_{2}}=\gamma_{k_{2}}^{\left(B\right)}{\mu (T)}^{k_{2}^{\prime }}$$
with dimensionless numerical constants $\gamma_{k_{1}}^{\left(A\right)}$ and $\gamma_{k_{2}}^{\left(B\right)}$. The gel condition becomes
(108)$$D\left(z_{A},z_{B}\right)=\left\{a-f^{\prime }{z_{A}}^{2}\phi_{2}^{\left(A\right)}\left(z_{A}\right)\right\}\left\{b-g^{\prime }{z_{B}}^{2}\phi_{2}^{\left(B\right)}\left(z_{B}\right)\right\}=0$$
with polynomials
(109a)$$\phi_{2}^{\left(A\right)}\left(z_{A}\right)\equiv \sum_{k_{1}\geq 2}k_{1}{k_{1}}^{\prime }\gamma_{k_{1}}^{\left(A\right)}{z_{A}}^{k_{1}^{\prime }}$$
(109b)$$\phi_{2}^{\left(B\right)}\left(z_{B}\right)\equiv \sum_{k_{2}\geq 2}k_{2}{k_{2}}^{\prime }\gamma_{k_{2}}^{\left(B\right)}{z_{B}}^{k_{2}^{\prime }}$$
and scaled concentrations
(110a)$$a=\lambda \left(T\right)\psi_{A}$$
(110b)$$b=\mu \left(T\right)\psi_{B}$$
and the scaled variables $z_{A},z_{B}$ instead of $\lambda z_{A},\mu z_{B}$. If the multiplicities $k_{1}$ and $k_{2}$ are fixed, the gel point condition, together with the materials conservation law, gives two solutions
(111a)$$a=\frac{f^{\prime }{k_{1}}^{\prime }}{{\left(\gamma^{\left(A\right)}\right)}^{1/{k_{1}}^{\prime }}{\left(f^{\prime }{k_{1}}^{\prime }-1\right)}^{k_{1}/{k_{1}}^{\prime }}}$$
(111b)$$b=\frac{g^{\prime }{k_{2}}^{\prime }}{{\left(\gamma^{\left(B\right)}\right)}^{1/{k_{2}}^{\prime }}{\left(g^{\prime }{k_{2}}^{\prime }-1\right)}^{k_{2}/{k_{2}}^{\prime }}}$$
corresponding to the SGT line of A-GEL and B-GEL. For pairwise case, we already reported on the thermoreversible IPN in detail [48,49]. An example IPN with multiple cross-linking is presented in Figure 8.

Let us briefly summarize the structure of binary gels after passing the gel pint. For binary gels, there are four regions in the post-gel regime: A-A connected region (referred to as A∪A) which has infinite paths on the gel connected by a homo A-A reaction, B-B connected region (B∪B) with infinite paths connected by a homo B-B reaction, A-B connected region (A∪B) with infinite paths connected by co-reaction only, and A-A/B-B connected region (A∩B) with infinite paths connected by both A-A reaction and B-B reaction: bi-continuous gel). The simplest example of A∩B is presented in Figure 8 for immiscible cross-links. The network is bi-continuous but separated with each other, and therefore it is called interpenetrating polymer network (IPN). More detailed analysis of the binary post-gel properties will soon be reported.

### 8.2. Completely Miscible Cross-Link Junctions

We next consider the opposite case where A- and B groups are similar and completely miscible with each other within a cross-link junction. Typical examples of such cases are associating polymers carrying many hydrophobes (large *f*) mixed with surfactant molecules (g=1) [35,56], and also mixed with telechelic polymers (g=2) (polymers carrying hydrophobes at both chain ends)[57]. If hydrophobes on both species are the same, they can completely mix with each other within miceller cross-link junctions formed by their aggregation [56].

For such a completely miscible junction, we can assume that $k_{1},k_{2}$ may vary under a total multiplicity $k_{1}+k_{2}=k$ which can also vary in a range fixed by the physical nature of interaction. We therefore have
(112)$$K_{k_{1},k_{2}}=\gamma_{k}\frac{k!}{k_{1}!k_{2}!}\lambda {\left(T\right)}^{k_{1}^{\prime }}\mu {\left(T\right)}^{k_{2}}\delta \left(k_{1}+k_{2}-k\right)$$
for the equilibrium constant for $k_{1}\geq 1$, where λ(T),μ(T) are the binding (association) constants of A- and B group, and $\gamma_{k}$ is a numerical constant. In what follows, we use the symbols $z_{A},z_{B}$ instead of $\lambda z_{A},\mu z_{B}$ for simplicity. The binding polynomial for A groups takes the form
(113)$$u_{A}\left(z_{A},z_{B}\right)=1+\sum_{k\geq 2}\gamma_{k}\sum_{k_{1}=1}^{k}\frac{k!}{k_{1}!k_{2}!}k_{1}{z_{A}}^{k_{1}^{\prime }}{z_{B}}^{k-k_{1}}=1+\sum_{k\geq 2}k\gamma_{k}{\left(z_{A}+z_{B}\right)}^{k^{\prime }}=\sum_{k\geq 1}k\gamma_{k}{\left(z_{A}+z_{B}\right)}^{k^{\prime }}$$

Similarly, we have
(114)$$u_{B}\left(z_{A},z_{B}\right)=\sum_{k\geq 1}k\gamma_{k}{\left(z_{A}+z_{B}\right)}^{k^{\prime }}$$
for B groups. Since $u_{A}=u_{B}$, we write it simply as $u(z_{A},z_{B})$. The materials conservation law is given by
(115a)$$a=z_{A}u\left(z_{A},z_{B}\right)$$
(115b)$$b=z_{B}u\left(z_{A},z_{B}\right)$$
in terms of $a\equiv \lambda \left(T\right)\psi_{A},b\equiv \mu \left(T\right)\psi_{B}$. Taking the sum, these coupled equations reduce to the one
(116)ψ=zu(z)
where ψ≡a+b, and $z\equiv z_{A}+z_{B}$. This relation is apparently the same as the materials conservation law for one component gels with multiple junctions studied in TS [44]. We have only to replace the concentration of the functional group by the weighted sum of the concentration of each component, and *z* variable by the sum of *z* for each component. By using the solution *z* of this equation, we can find $z_{A}=\left(1-w\right)z$ and $z_{B}=wz$, where
(117)w≡b/(a+b)
is the weighted number fraction of B groups among the total functional groups.

Now, from a simple calculation of $\hat{\kappa }$-matrix, we find the condition
(118)$$D\left(z\right)\equiv 1-[f^{\prime }\left(1-w\right)+g^{\prime }w]\kappa \left(z\right)=0$$
for SGT with one-component κ-function
(119)$$\kappa \left(z\right)\equiv \frac{d\ln u(z)}{d\ln z}$$

Because the prefactor $f^{\prime }\left(1-w\right)+g^{\prime }w$ can be regarded as weight-average functionality $f_{w}^{\prime }\equiv f_{w}-1$, where
(120)$$f_{w}\left(\xi \right)\equiv f\left(1-w\right)+gw=\frac{f+g\xi }{1+\xi }$$
the problem has entirely reduced to the gelation of one component functional molecules. In what follows, we refer to this fact as mixing law.

In order to study the existence of an optimal gelation point, we derive an additional condition dD(z)=0. Together with the relation dlnψ=[1+κ(z)]dz/z obtained from the materials conservation law, we find
(121)$$H_{B}\left(z\right)\equiv \kappa_{2}\left(z\right)\left(1+\xi \right)-\left(f^{\prime }-g^{\prime }\right)\kappa \left(z\right)\left[1+\kappa \left(z\right)\right]=0$$
for the appearance of the minimum in $\psi_{A}$ along SGT line, where a new $\kappa_{2}$-function is defined by the logarithmic derivative of κ(z)
(122)$$\kappa_{2}\left(z\right)\equiv \frac{d\ln\kappa (z)}{d\ln z}$$

By using u(z) only, we can write this condition explicitly as
(123)$$H_{B}\left(z\right)=g^{\prime \prime \prime }u\left(z\right)u^{\prime }{\left(z\right)}^{2}+g^{\prime }zu^{\prime }{\left(z\right)}^{3}+u{\left(z\right)}^{2}u^{\prime \prime }\left(z\right)=0$$
where abbreviated notations $g^{\prime \prime \prime }\equiv g-3,u^{\prime \prime }\left(z\right)\equiv d^{2}u\left(z\right)/dz^{2}$ have been used.

For the spinodal condition, we have
(124a)$$K_{A}=1+\left[w-f^{\prime }\left(1-w\right)\right]\kappa \left(z\right)$$
(124b)$$K_{B}=1+\left[1-w-g^{\prime }w\right]\kappa \left(z\right)$$
and
(125)$$D_{\mu }=1+\kappa \left(z\right)$$

#### 8.2.1. Pairwise Cross-Links

We first study the simplest case of pairwise cross-linking. We have only (2,0) (A${}_{2}$ cross-link), (1,1) (AB cross-link), and (0,2) (B${}_{2}$ cross-link). In general we have randomly mixed polymer networks if either of the functionality is larger than (or equal to) 3. This case was studied in detail in our previous paper [55]. If B molecule is monofunctional (g=1), it serves as a hinderer for gelation, i.e., gelation of A molecules is retarded, or prevented when B molecules are added to the solution (monotonically increasing $\psi_{A,c}$). If B molecule is bifunctional (g=2), generated networks have long AA subchains between neighboring cross-links [40]. In both cases, there is no optimal gel point.

#### 8.2.2. Fixed Multiplicity Model

We therefore move onto the multiple junctions. Let us first study a fixed multiplicity case of k(≥3), so that we have
(126)$$u\left(z\right)=1+\gamma kz^{k^{\prime }},\;\left(k^{\prime }\equiv k-1,\gamma \equiv \gamma_{k}\right)$$

From the gel point condition, we find
(127)$$\gamma k{z_{c}\left(\xi \right)}^{k^{\prime }}=1/\left[f_{w}^{\prime }\left(\xi \right)k^{\prime }-1\right]$$
and hence
(128)$$\lambda \psi_{A,c}=\frac{1}{{\left(\gamma k\right)}^{1/k^{\prime }}}\frac{f_{w}^{\prime }\left(\xi \right)k^{\prime }}{\left(1+\xi \right){\left[f_{w}^{\prime }\left(\xi \right)k^{\prime }-1\right]}^{k/k^{\prime }}}$$

Simple analysis of the condition (Equation 123) for optimal gel point leads to the conclusion that for monofunctional B molecules (g=1) it has a solution
(129)$$z^{*}={\left(\frac{k^{\prime \prime }}{\gamma k^{2}}\right)}^{1/k^{\prime }}$$

($k^{\prime \prime }\equiv k-2$) and therefore
(130)$$\xi^{*}\equiv \frac{\mu }{\lambda }R^{*}=\frac{1}{2}f^{\prime }k^{\prime \prime }-1$$
but for multifunctional B molecules (g≥2) there is no solution. One of the most interesting examples of such monofunctional B systems is the solutions of associating polymers mixed with surfactant molecules. There have been accumulated experimental data indicating the existence of a maximum in viscosity, and also in plateau modulus, as functions of the surfactant concentration [34,35]. Such maxima occur at the optimal concentration $\xi^{*}$ of surfactants for a given concentration of associating polymers.

#### 8.2.3. Mini-Max Model

Let us generalize the model to the multiplicity *k* allowed to vary in a certain interval between $k_{\min}$ and $k_{\max}$. We therefore have
(131)$$u\left(z\right)=1+\frac{d\phi (z)}{dz}$$
where
(132)$$\phi \left(z\right)\equiv \sum_{k=k_{\min}}^{k_{\max}}z^{k}=z^{k_{\min}}\frac{1-z^{\beta }}{1-z}$$
where $\beta \equiv k_{\max}-k_{\min}+1$. Gelation of such a mini-max junction was already studied for one-component networks [44]. We don’t repeat it here but instead focus on the appearance of the optimal gel point. Obviously g≤2 is necessary for the condition (Equation 123) to have a solution, but g=2 leads to a monotonic SGT line as studied above. Therefore B molecule must be a monofunctional one (g=1). A straightforward but lengthy calculation of $H_{B}\left(z\right)$ shows that the minimum multiplicity $k_{\min}$ must be larger than 3. If pairwise junction $k_{\min}=2$ is allowed, B molecules attach on the functional groups of A molecules to cover the surface, and prevent gelation. As a result, the gel concentration of A molecules monotonically increases. Speciality of $k_{\min}=2$ was already pointed out in our previous paper [56]. Upper bound in *k* has only weak effect so that the separation of cross-linked clusters into smaller pieces by the coverage of B molecules (monocomplexes) is not prevented by $k_{\max}$.

Figure 9 shows an example of the phase diagram with $k_{\min}=3,k_{\max}=5$ for a mixture of trifunctiona A molecules (f=3) and bifunctional B molecules (g=2). The association constant is fixed at λ=μ=2.0, while spinodal lines of two cases of the solvent $\chi_{A,B}=5.0,\chi_{A,0}=\chi_{B,0}=0$ (green broken line) and $\chi_{A,0}=0.9,\chi_{A,B}=\chi_{B,0}=0$ (blue broken line) are drawn.

Figure 10a shows phase diagram of telechelic associating polymers ($f=2,n_{A}=100$) mixed with surfactant molecules ($g=1,n_{B}=1$) with $k_{\min}=5,k_{\max}=20$. The association constant is fixed at λ=μ=80, while three spinodal lines for the solvent $\chi_{A,0}=0.60,0.603,0.7$ (poor for A molecules) with $\chi_{A,B}=\chi_{B,0}=0$ (blue broken lines) are drawn. Below $\chi_{A,0}=0.603$ (weakly poor solvent for the telechelic polymers), the two-phase region has a closed shape with a lower boud of B concentration. Because such closed-loop two-phase regions take place by adding cross-linking molecules which are perfectly miscible to the solvent, they are caused not by the enthalpy change but by the decrease in the mixing entropy of molecules due to association.

### 8.3. Partially Miscible Cross-Link Junctions

In mixtures of associating polymers and surfactants, the nature of the hydrophobes they carry may sometimes very different; with different species, different sizes, different shapes, etc. They are therefore only partially miscible within the miceller junctions. Another example of junctions with partial miscibility is formation of outer-sphere complexes in ion-binding cross-links. In this example, a metal ion forms a complex with fixed number of ligands ($\left(k_{1},1\right)$ type primary complex), and such primary complexes form higher order secondary complexes of the ${\left(k_{1},1\right)}_{k}$, where k=1,2,⋯. We can study thermoreversible gelation in such extended types of cross-link junctions with complex structure within the theoretical framework presented here. Some of the results on mixed micelles will be reported in our forthcoming paper.

## 9. Conclusions and Discussion

We have presented a very general theoretical framework for the study of binary thermoreversible gels in common solvents with specified multiple structures of mixed cross-links. All results are presented from a unified point of view in terms of the ternary phase diagrams of a new style by using scaled concentrations. The phase plane is different from that of the conventional triangular diagram; one axis of the plane is the concentration of the functional groups of the primary solute component (mainly polymers), and the other is the concentration of the functional groups of the secondary component (cross-linkers of various types) measured relative to that of the primary component. The method proposed is particularly convenient for the treatment of low-concentration region of the secondary component added to the solution.

The model solutions proposed in this study has obvious advantages in finding the microscopic parameters regarding the network junctions from macroscopic measurements of phase transitions. Hence, it can be regarded as a generalization of the conventional Eldridge–Ferry method [58,59] to binary multiple cross-links. From the theoretical and numerical results, together with qualitative comparison with experiments, the following conclusions can be drawn:There exists an optimal concentration of B (secondary) molecules for gelation if monocomplexes (surface coverage by B molecules) are formed at its high concentration region. The condition for optimization depends on the functionalities of both components, the multiplicity of the cross-links, and the association constants of complexes.In general, there are lower and upper bound in the concentration of B molecules between which the gel phase appears (reentrant sol-gel-sol transition). Both bounds depend on the functionalities of both components, the multiplicity of the cross-links, and the association constants of complexes.Gelation interferes with liquid–liquid phase separation. The relative position of the gel region and the two-phase region systematically shifts with the association constants of the cross-links and Flory’s interaction parameters. In one case, gelation takes place before phase separation (from clear gel to turbid gel), and in another vice versa (only turbid gel) with increase in the concentration of B molecules. The crossing point between the sol-gel transition line and the phase separation line is a higher order critical point.Addition of B molecules to the phase separated solutions of A molecules in a poor solvent may induce homogenization of the solution above a critical concentration and form a clear gel if the solvent is sufficiently good for B molecules. Oppositely, an already formed clear gel may turn into turbid one if the solvent is poor for B molecules.

Our theoretical framework may directly be applicable to some important thermoreversible gels, such as ion-binding PVA, galactomannan, etc. with borate, Ca${}^{2+}$, Fe${}^{3+}$ ions etc. The primary component in these studies are linear chains. Some of the results were already reported in our previous paper [26]. It may also be applicable to the problem of polymer-surfactant interaction. Solutions of associating polymers mixed with low molecular weight surfactants have been a focus of the researchers interest over the past decades. However, relationship between mixing property of the hydrophobes carried by them and their thermoreversible gelation has not been fully explored so far. Our model solution provides a good starting point to study the problem.

We have assumed that the equilibrium constants (K,λ,μ) for association are independent of Flory’s interaction parameters χ’s at our present stage of study. In the experiments they are often observed to depend on each other, because SGT of the same polymer takes place at different polymer concentrations in different solvents. In mixed solvents of different quality, in particular, there is a particular solvent composition at which gelation becomes easiest [60,61]. Study of the solvent effect will therefore require some more precise analysis of the structure of cross-link junctions in contact with solvent molecules.

Further important problem which remains open is the effect of temperature on the phase behavior. The main interest lies in the occurrence of UCST and LCST by association of solvent molecules [45,54]. A particularly important example is the association of water molecules onto polymer chains. In such water-soluble polymer solutions, the primary component is hydrogen-bonding polymer, the secondary component is an active solvent (ethanol, methanol, etc. or organic solvents), and the common solvent is water. Phase separation of such water-soluble polymers in mixed solvents have recently been studied in many researchers, but their gelation property remains as an open question.

Finally, we have to remark on the post-gel region. There are several treatments of the post-gel region in the phase diagrams in the theoretical study of polycondensation by tree statistics; one assumes tree structure for a gel network as for the sol, but the other permits cycle formation within the network. The former was proposed by Stockmayer [41], and the latter by Flory [37,38,39]. Later Ziff and Stell [62] examined another possibility from a kinetic point of view. There appear additional terms in the chemical potentials of each component depending on the treatment of sol-gel interaction in the post-gel region. For thermoreversible gels of one component with pairwise cross-linking in a solvent, these two conventional models were successfully applied without violating the thermodynamic requirements [50]. Therefore, the same treatments have to be also applied to the binary gels with multiple junctions. However, we have used in this study the same forms of the chemical potentials for the post-gel region as well as for the pre-gel region to avoid mathematical complexity. Detailed attempt to improve this point lies, however, outside of the scope of this study.

Generalization of the present theoretical method to multicomponent mixtures of functional molecules is straightforward. We have attempted its application to ternary polymer solutions in which all A,B,C functional groups are allowed to involve in a cross-link junction. It is hoped to pursue these topics in later publications.

## Figures and Tables

**Figure 1 gels-07-00089-f001:**
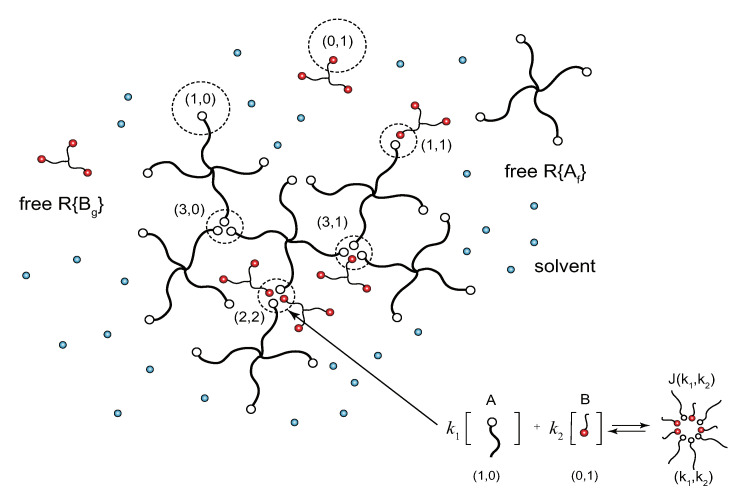
Connected tree-like clusters consisting of tetrafunctional molecules R{A${}_{4}$} and trifunctional molecules R{B${}_{3}$} in a solvent. Reversible chemical reaction in forming cross-link junctions contains arbitrary number $k_{1}$ of functional groups A and $k_{2}$ of groups B.

**Figure 2 gels-07-00089-f002:**
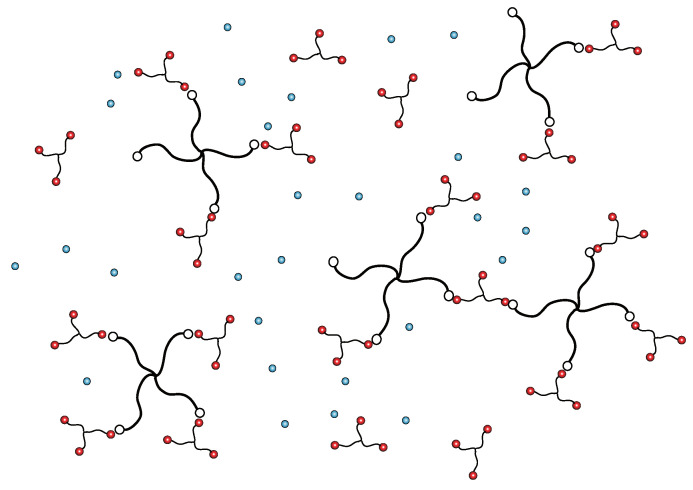
Reentrant sol region at high concentration of B molecules. Most of the clusters consist of small numbers of A molecules whose surfaces are covered by B molecules.

**Figure 3 gels-07-00089-f003:**
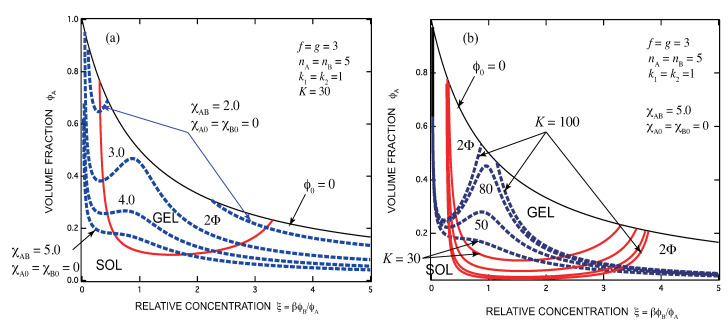
Sol-gel-sol transition lines (red solid lines) and spinodal lines (blue broken lines) for pairwise reaction $k_{1}=k_{2}=1$ drawn on the ternary phase plane. The horizontal axis is the scaled concentration ξ of the secondary functional groups B, and the vertical axis is the volume fraction $\phi_{A}$ of the primary functional groups A. The upper bound of the graph (thin solid line) shows the limiting solution without solvent. The functionalities are fixed at f=g=3. The number of repeat units are fixed at $n_{A}=n_{B}=5$. (**a**) The interaction parameter $\chi_{A,B}$ between A and B molecules is changed from 2.0 to 5.0 while the association constant is fixed at K=30. The two phase region is indicated by 2Φ. (**b**) The same as (**a**) but the association constant is changed from 30 to 100 while the interaction parameter is fixed at $\chi_{A,B}=5.0$.

**Figure 4 gels-07-00089-f004:**
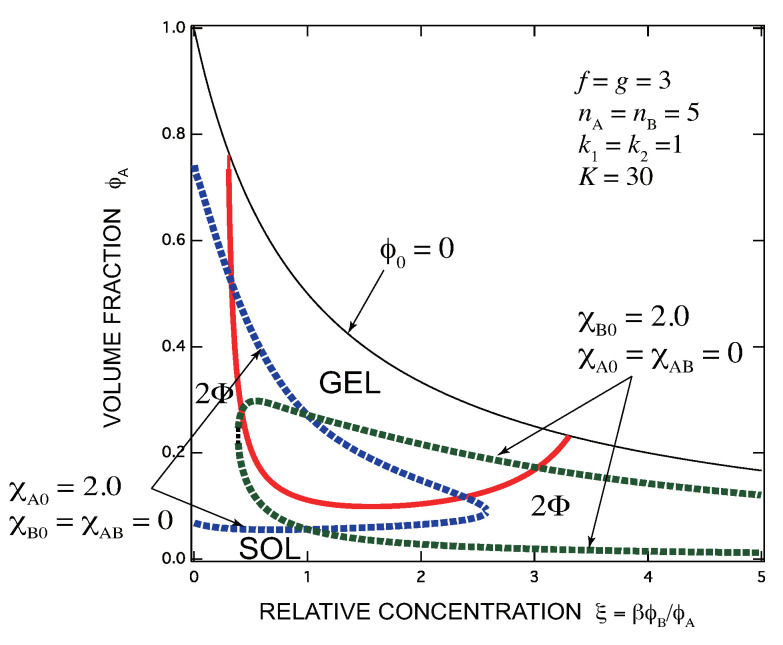
The same as Figure 3a but the interaction parameters are changed: (blue broken line) $\chi_{A,0}=2.0,\chi_{A,B}=\chi_{B,0}=0$, (green broken line) $\chi_{B,0}=2.0,\chi_{A,B}=\chi_{A,0}=0$. In the former case, the solution starts from a two-phase state, whose region shrinks with increase in the concentration of B molecules (good solvent for B), and eventually merges into one phase. Turbid gel turns into clear one with increase in B concentration. The latter case shows an opposite tendency; from clear gel to turbid gel.

**Figure 5 gels-07-00089-f005:**
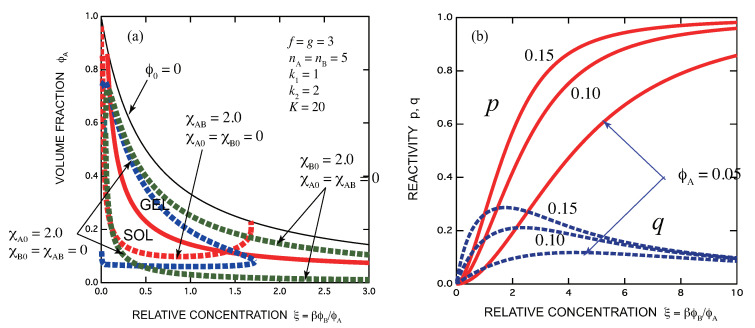
(**a**) Phase diagram of a mononuclear A cross-link junction; $f=g=3,\left(k_{1},k_{2}\right)=\left(1,2\right)$. The gel region is bound by $\xi_{\min}=0$ from below and $\xi_{\max}=g^{\prime }\left(f^{\prime }k_{2}+k_{2}^{\prime }\right)$ from above. SGST line (red line), spinodals for $\chi_{A,B}=2.0$ (red broken line), for $\chi_{A,0}=2.0$ (blue broken line), and for $\chi_{B,0}=2.0$ (green broken line) are shown. For the repulsive interaction $\chi_{A,B}=2.0$ in a good solvent $\chi_{A,0}=\chi_{B,0}=0$, phase separation takes place before the solution gels when the concentration of added B molecules is increased. (**b**) Reactivity *p* (solid lines) and *q* (broken lines) shown as functions of the scaled B concentration for three fixed A concentrations. They all start from 0. The reactivity of B groups shows a maximum near the stoichiometric concentration.

**Figure 6 gels-07-00089-f006:**
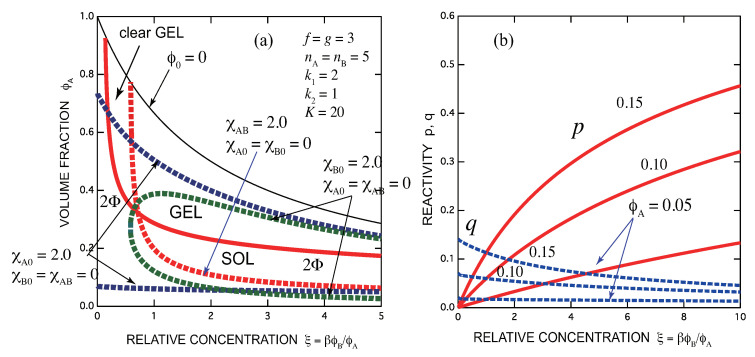
(**a**) Phase diagram for the mononuclear B cross-link junction. $f=g=3,\left(k_{1},k_{2}\right)=\left(2,1\right)$ (sandwich cross-link). The gel region is bound at $\xi_{\min}=1/f^{\prime }\left(g^{\prime }k_{1}+k_{1}^{\prime }\right)$ from below. There is no optimal concentration. SGT line (red line), spinodals for $\chi_{A,B}=2.0$ (red broken line), for $\chi_{A,0}=2.0$ (blue broken line), and for $\chi_{B,0}=2.0$ (green broken line) are shown. Adding trifunctional B molecules promotes gelation monotonically with B concentration. For the repulsive interaction $\chi_{A,B}=2.0$ in a good solvent $\chi_{A,0}=\chi_{B,0}=0$, gelation takes place before the solution phase separates when the concentration of added B molecules is increased. Therefore, there is a uniform clear gel region at low concentration of B molecules. (**b**) Reactivity *p* (solid lines) and *q* (broken lines) shown as functions of the scaled B concentration for three fixed A concentrations. They are monotonic functions of B concentration, but different from those of (1,2) type cross-links in that *q* starts from finite values.

**Figure 7 gels-07-00089-f007:**
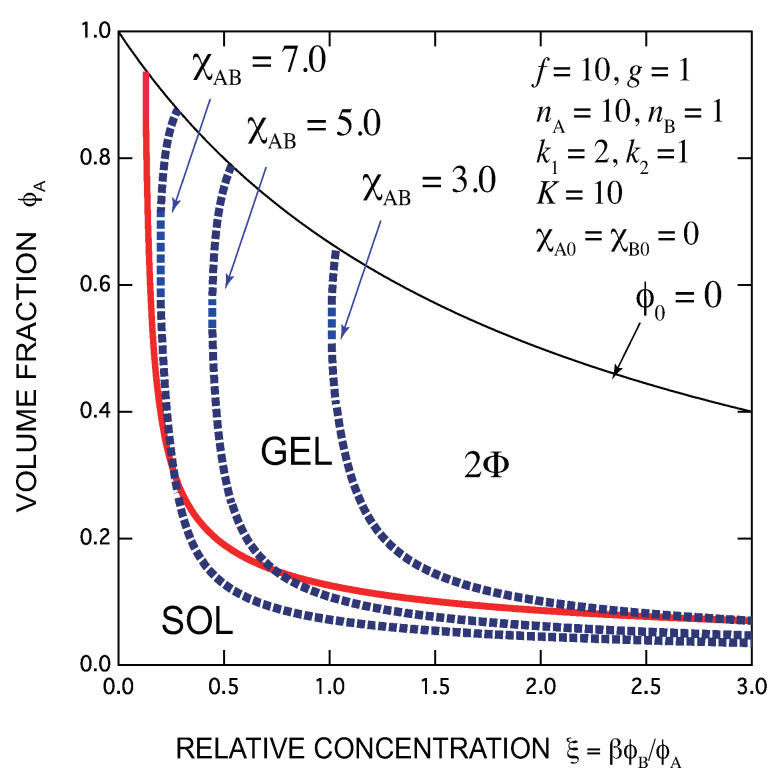
Phase diagram of monofunctional low molecular weight B molecules ($n_{B}=1,g=1$) with $\chi_{A,B}$ varied (blue broken line) in a good solvent for $f=n_{A}=10,\left(k_{1},k_{2}\right)=\left(2,1\right)$ (sandwich cross-link of longer molecules). The lower bound of the gel region is $\xi_{\min}=f^{\prime }k_{1}^{\prime }$. Many A functional groups remain active on the surface of clusters under excess of B molecules. Clear gel region shrinks with increase in $\chi_{A,B}$, and eventually disappears.

**Figure 8 gels-07-00089-f008:**
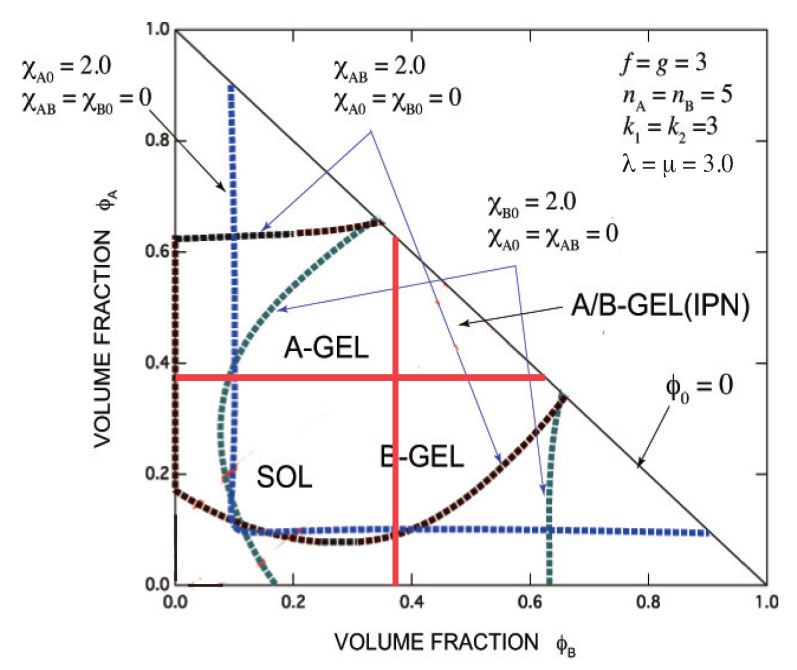
A typical example of phase diagrams for a symmetric IPN shown on the triangular volume fraction plane at a given temperature with fixed multiplicities $k_{1}=3,k_{2}=3$ for trifunctional A- and B molecules f=g=3 with λ=μ=3.0 and $\gamma^{\left(A\right)}=\gamma^{\left(B\right)}=1$. SOL region, A-GEL region, B-GEL region and IPN region (A/B-GEL) are indicated. Spinodal lines for AB interaction (blue broken line), for poor solvent of A (black broken lines) and for poor solvent of B (green broken lines) are also shown.

**Figure 9 gels-07-00089-f009:**
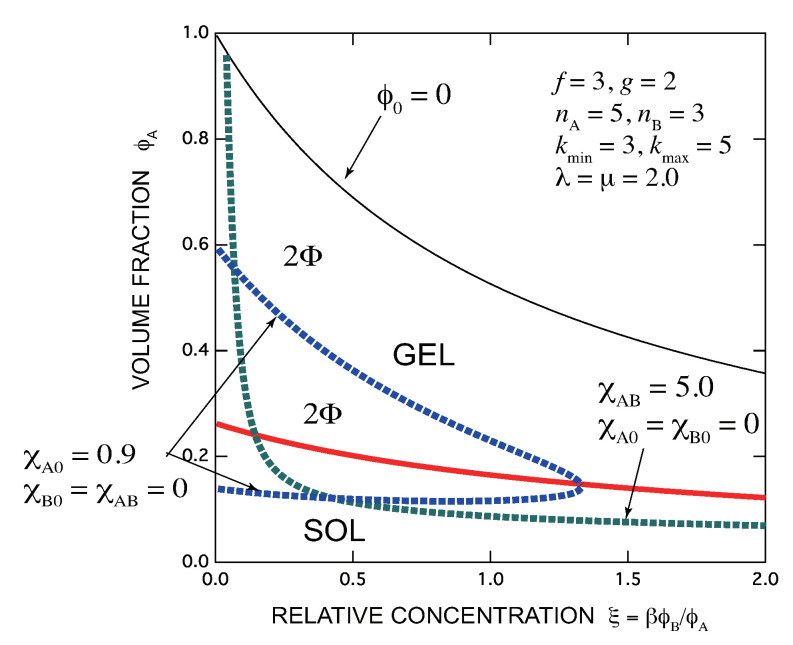
Phase diagram of a mixture of trifunctional A molecules and bifunctional B molecules in a solvent, whose functional groups are completely miscible in a range of multiplicity between 3 and 5. Solution of A molecules has SGT without B molecules, whose gel concentration monotonically decreases by mixing B molecules. Spinodal line for an interaction parameter $\chi_{A,B}=5.0$ between A- and B molecules in a good solvent (green broken line) and in a solvent poor to A molecules $\chi_{A,0}=0.9$ (blue broken line) are also drawn. For the latter, there is a critical B concentration above which the solution becomes uniform.

**Figure 10 gels-07-00089-f010:**
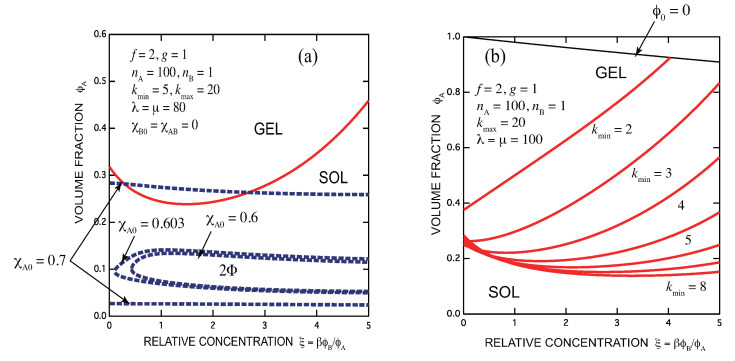
(**a**) Phase diagram of telechelic associating polymers ($f=2,n_{A}=100$) mixed with surfactant molecules ($g=1,n_{B}=1$). Hydrophobes carried by both molecules are assumed to be completely miscible within a formed micelles covering an allowed multiplicity range from $k_{\min}=5$ to $k_{\max}=20$. The polymer-solvent interaction parameter $\chi_{A,0}$ is changed for the calculation of spinodals (blue broken lines). The sol-gel transition (red line) has a minimum at a certain concentration of the surfactant near the stoichiometric one, so that it is an example of the reentrant sol-gel-sol transition. (**b**) The minimum multiplicity is allowed to change from 2 to 8 while the maximum multiplicity is fixed at 20. If $k_{\min}=2$ is allowed, the minimum concentration does not appear.

## Data Availability

The data presented in this study are available within the article.

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
