# Peer review of "Thermoreversible Gelation with Two-Component Mixed Cross-Link Junctions of Variable Multiplicity in Ternary Polymer Solutions"

_gels, 2021, doi:10.3390/gels7030089_

Round 1

Reviewer 1 Report

This manuscript reported on thermoreversible gelation with two-component mixed cross-link junctions of variable multiplicity in ternary polymer solutions. The authors described detailed models and their applications to thermoreversible gelation of binary reactive molecules in a common solvent. Interestingly, The structure and strength of network junctions were analyzed from macroscopic measurements of phase transitions. I think that this manuscript has provided interesting experimental results for the model solutions. And, overall experiments are systematically and well performed.

Reviewer 2 Report

Please find the attached PDF file.
